# Evidence for the nuclear import of histones H3.1 and H4 as monomers

Michael James Apta-Smith[1], Juan Ramon Hernandez-Fernaud[2] & Andrew James Bowman[1],*

## Abstract

**Newly synthesised histones are thought to dimerise in the cytosol and undergo nuclear import in complex with histone chaperones. Here, we provide evidence that human H3.1 and H4 are imported into the nucleus as monomers. Using a tether-and-release system to study the import dynamics of newly synthesised histones, we find that cytosolic H3.1 and H4 can be maintained as stable monomeric units. Cytosolically tethered histones are bound to importin-alpha proteins (predominantly IPO4), but not to histone-specific chaperones NASP, ASF1a, RbAp46 (RBBP7) or HAT1, which reside in the nucleus in interphase cells. Release of monomeric histones from their cytosolic tether results in rapid nuclear translocation, IPO4 dissociation and incorporation into chromatin at sites of replication. Quantitative analysis of histones bound to individual chaperones reveals an excess of H3 specifically associated with sNASP, suggesting that NASP maintains a soluble, monomeric pool of H3 within the nucleus and may act as a nuclear receptor for newly imported histone. In summary, we propose that histones H3 and H4 are rapidly imported as monomeric units, forming heterodimers in the nucleus rather than the cytosol.**

**Keywords** chaperone; chromatin; histone; nuclear import; nucleosome
**Subject Categories** Chromatin, Epigenetics, Genomics & Functional Genomics; Membrane & Intracellular Transport; Protein Biosynthesis & Quality Control
**The EMBO Journal (2018) 37: e98714**

## Introduction

Each cell division requires the doubling of both DNA and histone content, with half of the histones being of parental origin and half being newly synthesised. Whilst much effort has gone into studying the dynamics of recycled parental histones (Prior *et al*, 1980; Jackson, 1987, 1990; Katan-Khaykovich & Struhl, 2011; Radman-Livaja *et al*, 2011; Alabert *et al*, 2015), less is known about the program for newly synthesised histone incorporation. As they form the stable core of the nucleosome and are the substrates for the majority of post-translational marks, histones H3 and H4 are often at the forefront of these investigations.

Unlike recycled histones, newly synthesised histones H3 and H4 must pass through the cytosol before they are incorporated into chromatin. Biochemical isolation of H3.1 (the replication-dependent H3 variant) containing complexes suggests it folds with H4 soon after synthesis, interacting with a number of histone chaperones to form a cytosolic chaperoning network that coordinates nuclear import (Mosammaparast *et al*, 2002; Campos *et al*, 2010; Alvarez *et al*, 2011; Ask *et al*, 2012). Key cytosolic events in the proposed pathway include H3.1 and H4 forming a heterodimer in the cytosol and interacting with NASP, ASF1, HAT1 and RbAp46 (Mosammaparast *et al*, 2002; Campos *et al*, 2010; Alvarez *et al*, 2011), HAT1 modification of H4 K5 and K12 by acetylation (Alvarez *et al*, 2011; Parthun, 2011), modification of H3 K9 by methylation (Pinheiro *et al*, 2012; Rivera *et al*, 2015) and association with the importin-β protein IPO4 (Imp4b) (Mosammaparast *et al*, 2002; Blackwell *et al*, 2007; Campos *et al*, 2010; Ask *et al*, 2012; Keck & Pemberton, 2012; Gurard-Levin *et al*, 2014; Hammond *et al*, 2017). In addition, a number of importin-β proteins have been suggested to provide chaperoning roles for basic nuclear cargo including ribosomal proteins and linker histones (Jakel *et al*, 2002), but not, as yet, the core histones.

Nuclear import and incorporation of histones into chromatin occurs very rapidly (Ruiz-Carrillo *et al*, 1975; Bonner *et al*, 1988), thus following such events in a pulse-chase manner remains challenging. Import rates most likely exceed the folding and maturation kinetics of fluorescent proteins, making the process difficult to study by FRAP, FLIP, photoactivation or their derivative techniques (Reits & Neefjes, 2001; Lukyanov *et al*, 2005; Ishikawa-Ankerhold *et al*, 2012). Similar difficulties arise with self-labelling domains such as the SNAP-tag, requiring minutes to hours for quenching, pulsing and labelling steps (Juillerat *et al*, 2003; Jansen *et al*, 2007; Crivat & Taraska, 2012; Clement *et al*, 2016). Metabolic incorporation of radioactive amino acids or functional amino acid derivatives (Dieterich *et al*, 2007; Deal *et al*, 2010; Lang & Chin, 2014) affords immediate labelling for biochemical analysis, but presents challenges for imaging proteins due to the requirement for derivatisation of the incorporated functional groups (Lang & Chin, 2014), and are subject to artefacts arising from biochemical purification processes. Thus, many of the current ideas about the nucleo-cytoplasmic chaperoning of histones remain to be tested in a cellular setting.

1 Division of Biomedical Sciences, Warwick Medical School, University of Warwick, Coventry, UK
2 Proteomics Research Technology Platform, School of Life Sciences, University of Warwick, Coventry, UK
*Corresponding author. Tel: +44 2476 150220; E-mail: a.bowman.1@warwick.ac.uk

In an attempt to address this, and potentially gain new information regarding the histone import and deposition pathway, we have developed an approach termed RAPID-release (rapamycin activated protease through induced dimerisation and release of tethered cargo) that allows observation of dynamic cellular events in real time in living cells. In this approach, we circumvent the requirement for immediate labelling of newly synthesised histones by first capturing them on the cytosolic face of the outer mitochondrial membrane (OMM). The quiescent histones are then released by concomitant recruitment and activation of a site-specific, viral protease through the addition of the small molecule rapamycin (Stein & Alexandrov, 2014). The histones can be followed by fusion of a fluorescent protein, allowing visualisation of nuclear import and incorporation at replication domains in real time. We apply this approach to investigate the early maturation and import of H3.1 and H4, corroborating our findings through quantitative analysis of histone stoichiometries bound to core chaperoning components.

## Results

### Mislocalisation of histone chaperones during fractionation necessitates an *in vivo* pulse-labelling system

During analyses of histone chaperone localisation, we observed a striking discrepancy between biochemical fractionation and immunofluorescence in the sub-cellular localisation of a number of histone chaperones. Whilst s/tNASP, HAT1, RbAp46 and ASF1A appear overwhelmingly cytosolic using a standard NP-40 lysis protocol (Suzuki *et al*, 2010), with CAF1p60 almost equally split, they appear entirely nuclear when probed by immunofluorescence (compare Fig 1A with B). Pre-blocking of the antibodies with their immunogens specifically reduced the nuclear fluorescence, suggesting the antibodies are specific and the discrepancy is not due to cross-reactivity (Fig 1C, example images in Fig EV1A). Furthermore, the chaperones did not change their localisation between S and G1/G2 phases, as determined by the presence or absence of PCNA foci in the nucleus (Fig EV1B), suggesting the discrepancy cannot be due to cell cycle effects.

We reasoned one explanation for the discrepancy could be the leakage of nuclear components into the cytosolic fraction during biochemical isolation. To determine the behaviour of the soluble nuclear chaperones during fractionation, we took mCherry-sNASP as a representative nuclear component of the histone chaperoning pathway and co-expressed it in HeLa cells with EGFP-Lamin A/C to act as a marker of nuclear integrity. Addition of cytosolic extraction buffer (PBS + 0.1% NP-40) caused nuclei to puff-up (Fig 1D) but remain in close proximity to their pre-lysis position allowing us to follow mCherry-sNASP's location. Interestingly, whilst Lamin A/C remained at the nuclear periphery throughout the time course, sNASP rapidly diffused out of the nucleus (Fig 1D and E, Movie EV1). Similar behaviour was seen for hypotonic lysis in which cells were monitored whilst being exposed to water (Fig EV1C). It should be noted that previous studies using *Xenopus* oocytes have recorded a similar nuclear leakage with regard to total protein levels (Paine *et al*, 1983, 1992).

In summary, we propose that the discrepancy between biochemical observations and *in situ* observations may be explained in terms of the rapid diffusion of soluble nuclear components during separation of cytoplasmic and nuclear compartments, and that the core histone chaperoning proteins NASP, ASF1A, RbAp46 and HAT1 reside predominantly in the nucleus throughout the cell cycle.

### The RAPID-release technique allows observation of nascent cytosolic histones and their translocation to the nucleus

Taking into account the problem of nuclear leakage detailed above, we wondered whether it is possible to probe cytosolic histones in living cells using fluorescence microscopy. Whilst fluorescent microscopy has been used extensively to study histone turnover in chromatin, the kinetics with which histones are incorporated into chromatin after synthesis (Ruiz-Carrillo *et al*, 1975; Bonner *et al*, 1988) is likely an order of magnitude greater than that of fluorescent protein folding or SNAP/HALO-tag labelling, and thus a histone will be incorporated into chromatin before it is observed. To circumvent this, we pursued a cytosolic tether-and-release strategy termed RAPID-release (rapamycin activated protease through induced dimerisation and release of tethered cargo). In this approach, histones are tethered to the cytosolic face of the outer mitochondrial membrane (OMM) and are held in a quiescent state whilst the fluorescent fusion protein matures. Detethering is triggered by addition of rapamycin, which concomitantly recruits and activates an auto-inhibited TVMV protease (Stein & Alexandrov, 2014), thus allowing observation of nuclear import and chromatin deposition.

To assess the feasibility of the approach, we fused EGFP to the FKBP12-rapamycin-binding (FRB) domain of mTOR, followed by the mitochondrial tail-anchoring sequence of OMP25 (Horie *et al*, 2002), with two TVMV cleavage sites separating the EGFP and FRB-OMP25 domains (EGFP$^{TVMVx2}$-FRB-OMP25) (Fig 2A). The TVMV protease containing a C-terminal auto-inhibitory (AI) peptide and an N-terminal FK506-binding domain (FKBP12) (Stein & Alexandrov, 2014) was fused to the C-terminus of mCherry, creating the construct mCherry-FKBP12-TVMV-AI (Fig 2A). Addition of rapamycin to HeLa cells co-transfected with EGFP$^{TVMVx2}$-FRB-OMP25/mCherry-FKBP12-TVMV-AI resulted in the recruitment of mCherry-FKBP12-TVMV-AI to mitochondria and release of the EGFP cargo from its tether (Fig 2B). Analysis of the cleavage (as the change in maximum pixel intensity of the cytoplasm) revealed a fit to an exponential decay model (Fig 2C). Plotting the rate constant of fit against the relative expression level of the protease (measured as the ratio of mCherry:EGFP signal) for each cell revealed a strong positive correlation, suggesting the cleavage rate is dependent on the level of protease (Fig 2D). At the highest ratios of protease to substrate, a half-maximal cleavage of 2.5 min was achieved. Removal of the two TVMV cleavage sites from EGFP inhibited cleavage, whilst removal of the AI peptide from the TVMV protease resulted in constitutive activity without recruitment (Fig EV2A and B), demonstrating the specificity of the protease and the importance of the AI peptide fusion, respectively.

Next, we used the RAPID-release system to observe the dynamics of histone H3.1 and H4 upon release from the OMM. In this instance, tail anchoring (Suzuki *et al*, 2002), in contrast to N-terminal anchoring, permitted C-terminal tagging of histones, allowing us to avoid N-terminal fusions that have previously been shown to affect chromatin incorporation dynamics (Kimura & Cook, 2001). Transient transfection of H3.1 or H4 fused to the N-terminus of the EGFP$^{TVMVx2}$-FRB-OMP25 construct did not observably affect

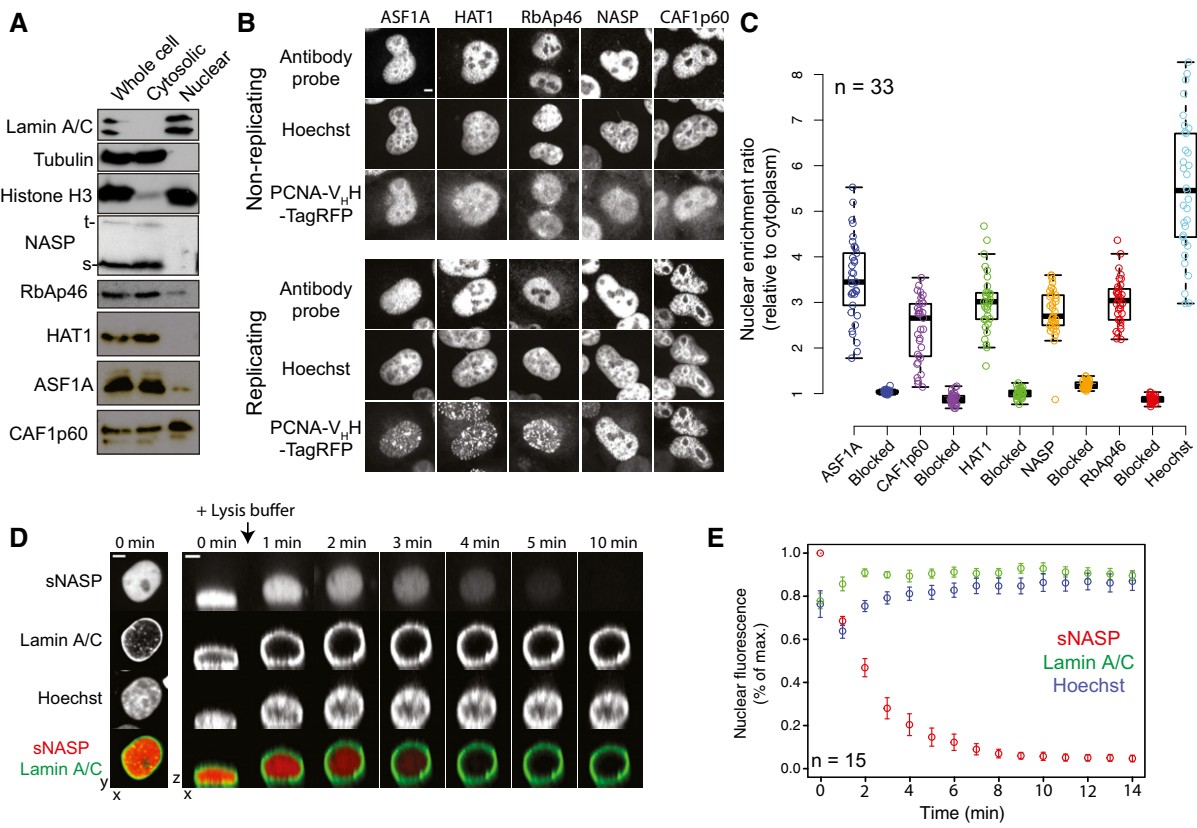

**Figure 1.  Sub-cellular localisation of core H3.1 and H4 chaperoning components.**

A   Fractionation of nuclear and cytosolic compartments using a standard NP-40 lysis protocol and immunoblotting with chaperone specific antibodies.
B   Immunofluorescence of histone chaperones in fixed cells using the same antibodies used in (A). Cells are segregated into those undergoing replication and those that are not. Scale bar represents 5 μm.
C   Quantification of nuclear localisation shown in (B). Boxes represent the lower quartile, median and upper quartile. Whiskers represent 1.5 times the interquartile range.
D   Real-time imaging of cells undergoing biochemical fractionation. sNASP is tagged with mCherry, whilst Lamin A/C is tagged with EGFP. The left column of panels represents a maximum intensity projection. The time course on the right represents a plane in the z dimension reconstructed from 20 z-stacks penetrating 20 μm into the medium. Scale bars represent 5 μm.
E   Quantification of nuclear leakage shown in (D). Z-stacks were flattened using a maximum intensity projection with the nuclear fluorescent signal over time plotted as a percentage of maximum (normalised). Data points represent the mean of 15 measurements with error bars representing the SEM.

Source data are available online for this figure.

mitochondrial or cellular morphology. However, a small amount of background nuclear fluorescence was observed (Fig 2E, Movies EV2 and EV3). Release of H3.1/H4-EGFP from the cytosolic tether resulted in rapid nuclear localisation (measured as the median nuclear fluorescence) (Fig 2E and F) at a rate mirrored by the kinetics of cleavage (measured as the S.D of the cytoplasm) (Fig 2G). Standard deviation of the cytoplasm was used instead of maximum pixel intensity as it was less affected by sub-cellular partitioning. Plotting the nuclear import rate against the cleavage rate resulted in a strong fit to a linear model (Fig 2G), revealing nuclear import of histones occurs at a rate greater than proteolytic cleavage and in excess of our sampling rate. Confirming this, the modal value of the partitioned cytoplasmic signal, representing the portion of the cytoplasm outside of the mitochondrial network, did not increase over the cleavage period (Fig EV2C) as it did for freely diffusing EGFP.

In summary, the RAPID-release technique allows observation of histone nuclear import in living cells and provides a pulse-chase strategy with significantly improved kinetics compared to currently available techniques.

**Histones released from their tether incorporate rapidly at actively replicating domains**

In order to validate the tether-and-release approach in studying histone deposition, we tested cells for their ability to incorporate released histones into their chromatin. Mammalian genomes are organised into topological domains (TADs), which have been suggested to correlate with stable units of replication, or replication domains (RDs) (Pope *et al*, 2014; Rivera-Mulia & Gilbert, 2016). In a subset of asynchronously dividing cells, we observed foci forming in the nucleus soon after histone release (Fig 3A). To determine whether these foci represent histone incorporation at RDs, we co-transfected H3.1-EGFP$^{TVMVx2}$-FRB-OMP25 and TagBFP-FKPB-TVMV-AI fusions with a PCNA-V$_H$H-TagRFP chromobody to mark

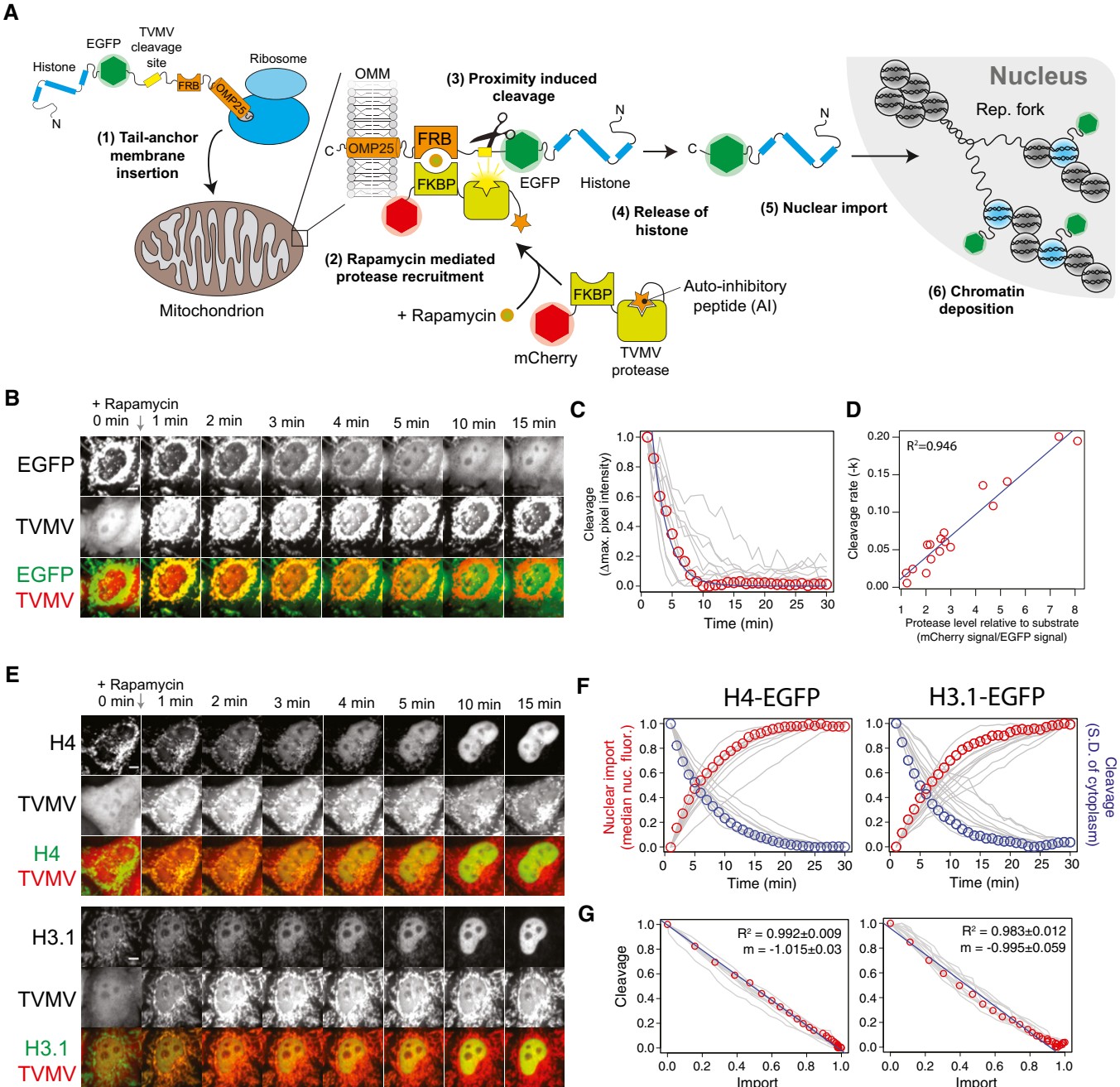

**Figure 2.  A tether-and-release approach for studying histone dynamics in living cells.**

A   Schematic representation of the RAPID-release technique. Addition of rapamycin results in recruitment and activation of an auto-inhibited TVMV protease, leading to the release of EGFP-labelled histones from their mitochondrial tether. TVMV—tobacco vein mottling virus, FKBP—FK506-binding protein (FKBP12), FRB—FKBP12-rapamycin-binding domain, OMM—outer mitochondrial membrane, OMP25—25 kDa outer-membrane immunogenic protein (C-terminal helix), AI—auto-inhibitory peptide.

B   A representative cell showing release of tethered EGFP. Scale bar represents 5 μm.

C   Quantification of EGFP release. Single cells are represented as grey traces. The cell displayed in (B) is highlighted with red circles. The blue line represents a fit to an exponential decay model.

D   Rate constants from exponential decay functions of the traces shown in (C) plotted against the expression level of protease relative to EGFP substrate. The blue line represents a linear regression model with the $R^2$ value shown.

E   Representative cells showing release of H4-EGFP and H3.1-EGFP. Scale bar represents 5 μm.

F   Nuclear import rate and cleavage rate of H3.1-EGFP and H4-EGFP. Single cells are represented as grey traces. Traces from cells displayed in (E) are highlighted with circles.

G   Import rate plotted against cleavage rate. Values represent those shown in (F), with the cells shown in (E) highlighted with red circles. Linear regression models are shown with blue lines. The mean $R^2$ and slope (m) are shown for the 11 cells quantified in (F), +/− relates to standard deviation.

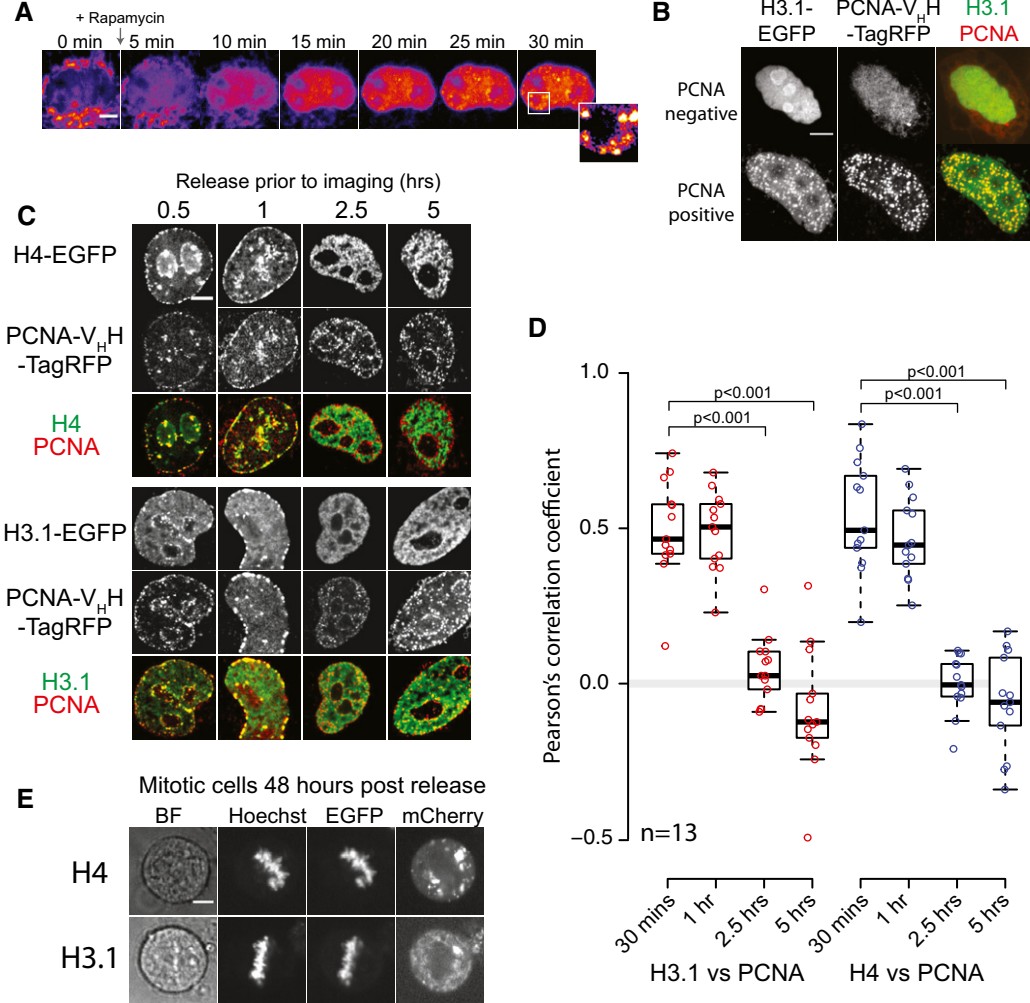

**Figure 3.  Incorporation of released histones into chromatin at sites of active replication.**

A   Accumulation of nuclear H3.1-EGFP and the formation of distinct foci. Scale bar represents 5 μm.

B   Representative cells showing two different subpopulations of released H3.1-EGFP staining and their colocalisation with PCNA-positive foci. PCNA was detected with a PCNA-V$_H$H-TagRFP chromobody. Scale bar represents 5 μm.

C   Representative cells showing points from a time course of H3.1-EGFP and H4-EGFP release. Scale bar represents 5 μm.

D   Colocalisation of PCNA and H3.1/H4-EGFP signals at various time points after release. A Wilcoxon rank sum test was carried out for each time point versus the 30-min time point, with those scoring a *P*-value of < 0.001 indicated by brackets.

E   Released H3.1/H4-EGFP colocalise with mitotic chromosomes 48 h post-release. Scale bar represents 5 μm.

active replication (Burgess *et al*, 2012) (the TagBFP channel was used to identify expressing cells, but was not imaged to minimise bleaching of the other fluorophores). Fixing and imaging cells 30 min after rapamycin addition showed that nuclei positive for PCNA foci were also positive for H3.1-EGFP foci, whereas nuclei negative for PCNA foci were also negative for H3.1-EGFP foci (Fig 3B). Cells absent of PCNA foci, and by deduction not replicating their genomes, demonstrated a nucleolar enrichment of H3.1-EGFP, which has previously been shown to be an artefact of excess soluble histones in the nucleus (Musinova *et al*, 2011; Safina *et al*, 2017).

To analyse the incorporation dynamics more quantitatively, we released histones at four time points prior to fixation and colocalisation analysis. Cells in mid/late S-phase were chosen, identified from their peripheral PCNA staining pattern (Burgess *et al*, 2012), as these cells would have been in S-phase as histones were released. Within 30 min, formation of foci occurred in the EGFP channel that colocalised to replication domains marked by the PCNA chromobody (Fig 3C and D). Colocalisation continued for 1 h, then decreased dramatically at 2.5 h, with a slight anti-correlation seen at 5 h post-release (Fig 3C and D). We interpret this as the pulse of released histones entering the soluble pool and being incorporated at actively replicating domains, up until the fluorescent pool of histones is depleted and replication moves on to neighbouring domains.

To further verify that released histones are incorporated into chromatin, we imaged cells in mitosis 48 h post-release and found the H3.1-/H4-EGFP signal localised to the condensed, mitotic chromosomes, providing further evidence that released H3.1-/H4-EGFP are deposited into chromatin and stably retained through cell division (Fig 3E). Together, these experiments demonstrate that

released H3.1-/H4-EGFP enter the histone chaperoning pathway and are incorporated into chromatin in a similar fashion to endogenous histones, and validate the RAPID-release system as a method for investigating chromatin assembly and the histone chaperoning pathway.

## Cytosolically tethered H3.1 and H4 are monomeric and do not detectibly associate with endogenous NASP, ASF1A, RbAp46 or HAT1

H3.1 and H4 exist as an obligate heterodimer in chromatin (Luger *et al*, 1997) and are also found as a dimer when bound to a number of histone chaperoning proteins, such as ASF1A/B, s/tNASP, RbAp46, HAT1 and the CAF1 complex (Tagami *et al*, 2004). However, the two histones are synthesised separately and must fold at a point prior to entry into the chromatin deposition pathway. We reasoned that if H3.1 and H4 fold in the cytoplasm, we would expect to see enrichment of the endogenous partner on the mitochondrial network. Similarly, endogenous histone chaperones that interact with H3.1 and H4 should also be enriched at the mitochondrial network (Fig 4A). To test this, we carried out immunofluorescence to probe for the co-occurrence of endogenous histone binding partners (Figs 4B and EV3A and B). Pearson's correlation coefficients of cytosolic regions encompassing the mitochondrial network were calculated for the EGFP and immunofluorescent channels (Figs 4C and EV3A and B). Interestingly, whilst we could detect the tethered histones, we could not detect enrichment of their orthologous binding partners, nor could we detect any of the histone chaperones (Fig 4C), most likely due to their nuclear localisation (Figs 1 and EV3B).

To determine whether lack of binding was due to the nuclear partitioning of histone chaperones, rather than tethered histones adopting an unfavourable conformation that prevents binding, we expressed forced cytosolic chaperones that were mCherry-tagged. To achieve cytosolic localisation, chaperones were either mutated in their nuclear localisation sequence (ΔNLS), where a defined NLS existed (as for NASP; Kleinschmidt & Seiter, 1988; O'Rand *et al*, 1992), or engineered with a strong nuclear export signal (NES) (Henderson & Eleftheriou, 2000) where no known NLS was present (as for RbAp46, HAT1, ASF1A) (Fig 4D). This effectively drove the cytosolic location of all histone chaperones tested (Fig EV3C). The rationale behind the experiment was as follows: as RbAp46 and HAT1 bind to H4 epitopes within the H3.1-H4 heterodimer (Murzina *et al*, 2008; Song *et al*, 2008) (Fig 4G), if tethered histones were

folded with their endogenous counterpart, we would expect to see the recruitment of RbAp46 and HAT1 to both tethered H3.1 and tethered H4, whereas if histones were monomeric, we would expect to see recruitment to tethered H4, but not H3.1. Conversely, as sNASP interacts directly with H3 as a monomer and as an H3.1-H4 heterodimer (Bowman *et al*, 2016, 2017), we would expect to see recruitment to both tethered H3.1 and H4 if a heterodimer was present, but only to tethered H3.1 if the histones were monomeric. As ASF1 contacts both H3 and H4 through independent binding sites (English *et al*, 2005; Natsume *et al*, 2007), we may expect to see recruitment to both independent of the histone's oligomeric status. Interestingly, forced cytosolic RbAp46-NES and HAT1-NES both localised to tethered H4 but not H3.1, whereas sNASP-ΔNLS localised to tethered H3.1 but not H4 (Figs 4F and EV3C). Due to having distinct binding sites for each histone (Fig 4G), ASF1A interacted with both H3.1 and, to a lesser extent, H4.

Taking into consideration the inability to detect the endogenous histone counterparts of tethered H3.1 and H4, and the binding profiles of the forced cytosolic chaperones, our results suggest that the majority of tethered histones reside in their monomeric form, not associating with endogenous ASF1A, s/tNASP, RbAp46 or HAT1, which appear to be predominantly nuclear in location.

## The importin-β nuclear receptors stably interact with cytosolic H3.1 and H4

Previous reports suggested importins may play a dual role in both chaperoning basic nuclear proteins and acting as receptors in delivering them to the nucleus (Jakel *et al*, 2002). Indeed, human IPO4 (also known as Imp4, Imp4b and RanBP4) has been isolated bound to H3.1 and H4 in complex with ASF1 from HeLa cell extracts (Campos *et al*, 2010; Jasencakova *et al*, 2010; Ask *et al*, 2012). To further investigate whether importins interact with our cytosolically tethered histones, we screened three importin-β family members—IPO4, KPNB1 (IMB1, PTAC97, NTF97) and IPO11 (Imp11, RanBP11) for colocalisation to the mitochondrial network (Fig 5A and B). Based on phylogenetic analysis of the importin-β family (O'Reilly *et al*, 2011), IPO4 and KPNB1 were chosen as representatives of two closely related branches of the import exclusive importin-β superfamily, whereas IPO11 was chosen as a member of a more distantly related branch (Fig 5C). We screened a number of antibodies to these importin, but could not find any that were suitable for immunofluorescence, and so assessed interaction through colocalisation of mCherry-importin fusions. Interestingly, we found that

---

**Figure 4.   Interaction profiling of tethered cytosolic histones.**

A   Schematic representation of the fluorescence-2-hybrid approach for analysing interaction with endogenous proteins.

B   Example of interaction screening using an α-H3 antibody against tethered EGFP (EGFP-OMP25), tethered H3.1 (H3.1-EGFP-OMP25) and tethered H4 (H4-EGFP-OMP25). Background-corrected, single Z-slices of representative cells are shown with 2D histograms displayed on the right. Nucleus and cytoplasm are portioned with a white line. Pearson's coefficients (*R*) for the cytosolic region of the depicted cells are shown in the histogram inset. Scale bar represents 10 μm.

C   A boxplot of Pearson's coefficients between tethered EGFP, H3.1-EGFP or H4-EGFP and endogenous histone counterparts or known histone chaperones. *P*-values of < 0.001 from a Wilcoxon rank sum test versus the EGFP alone control are indicated by brackets. Whiskers extend to 1.5 times the IQR.

D   Schematic representation of a fluorescence-2-hybrid assay using mCherry-tagged, forced cytosolic chaperone.

E   Example of interaction screening as in (B), but using sNASP-dNLS against EGFP (EGFP-OMP25), tethered H3.1 (H3.1-EGFP-OMP25) and tethered H4 (H4-EGFP-OMP25). Scale bar represents 10 μm.

F   A boxplot of Pearson's coefficients between tethered EGFP, H3.1-EGFP or H4-EGFP and cytosolically forced chaperones. *P*-values of < 0.001 from a Wilcoxon rank sum test versus the EGFP control are indicated by brackets. Whiskers extend to 1.5 times the IQR.

G   Crystal structures of RbAp46-H4, HAT1-H4 and ASF1A-H3-H4 complexes. H3 is shown in red, and H4 is shown in blue.

---

                    

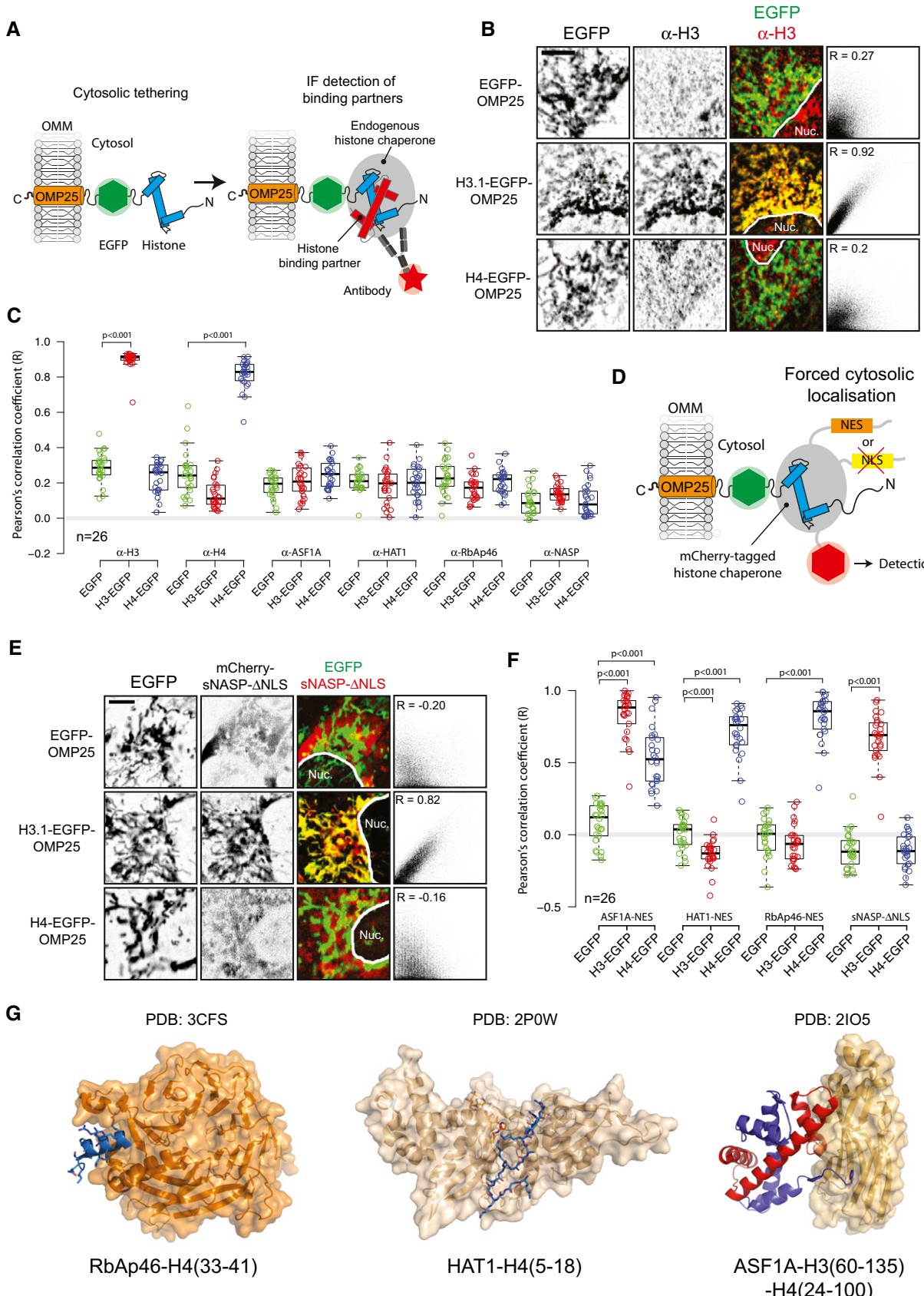

**Figure 4.**

IPO4 and KPNB1 interacted with tethered H4, whereas IPO4, but not KPNB1, interacted with tethered H3.1 (Fig 5A and B). IPO11 did not interact with either tethered H3.1 or H4 (Fig 5A). mCherry on its own served as a control for non-specific binding. Recent crystal structures have identified the importin-β protein TNPO1 (KPNB2, MIP1), closely related to KPNB1 (O'Reilly *et al*, 2011), as interacting with the H3 tail (Fig EV4) (Soniat & Chook, 2016), and the *Kluyveromyces lactis* Kap123, a homolog of IPO4, as interacting with both H3 and H4 tails (An *et al*, 2017), building on previous biochemical and genetic analysis (Mosammaparast *et al*, 2002; Blackwell *et al*, 2007; Soniat & Chook, 2015).

The histone tail regions of H3 and H4 have been previously identified as nuclear localisation signals (Mosammaparast *et al*, 2001, 2002; Blackwell *et al*, 2007); however, nuclear import of histones in these studies has used trans-genes expressed from constitutively active promoters. To investigate the regions of H3 and H4 important for import using the RAPID-release approach, we generated truncations of H3.1 and H4 and tested their ability to be imported at the course of 30 min after release (Fig EV4C and D). Interestingly, we find that the junctions between the tail regions and the globular domains in both H3 and H4 are important for the efficient nuclear import of histones in human cells. For H3, residues 1-20 showed no enrichment, whereas residues 1-30 showed only a partial enrichment in the nucleus, whilst a region encompassing the α-N helix was necessary for full localisation to the nucleus (Fig EV4C and D). Similarly, for H4, the shorter tail domain (residues 1–20) was not sufficient for nuclear import, but a fragment including the α1 helix of H4 was necessary for efficient nuclear localisation (Fig EV4C and D).

To probe the histone-IPO4 interaction further, we used fluorescence recovery after photobleaching (FRAP) to assess the turnover of soluble mCherry-IPO4 on tethered histones. A region of the cytosol in cells co-expressing tethered H3.1/H4-EGFP and mCherry-IPO4 was bleached using a 561 nm laser, with recovery monitored for 20 min post-bleaching. For comparison, turnover of the artificially localised mCherry-sNASP-ΔNLS and mCherry-RbAp46-NES was also measured for H3.1 and H4, respectively. Remarkably, we observed very little turnover of mCherry-IPO4 on the mitochondrial network, whereas binding by the forced cytosolic chaperones was highly dynamic, recovering almost immediately (Fig 5D and E). This suggests that monomeric histones present a stable substrate for importin binding, but not for the nuclear chaperones NASP and RbAp46.

Next, we asked what happens when the histones are released from their cytosolic tether. H3.1-EGFP$^{\text{TVMVx2}}$-FRB-OMP25, mCherry-IPO4 and TagBFP-FKBP12-TVMV-AI were co-transfected into HeLa cells

and imaged every minute after rapamycin-induced histone release. As H3.1-EGFP was released from its mitochondrial tether, IPO4 diffused away from the mitochondrial network, as expected, but in contrast to H3.1-EGFP, remained cytosolic in its location (Figs 5F–H and EV4A, Movie EV4). One may have expected a transient increase in nuclear IPO4 levels after H3.1 release as the IPO4-H3.1 complex translocates to the nucleus. However, this would require a sampling time in excess of nuclear import rate. Limited by the kinetics of histone release, our imaging rate is most likely in dearth of nuclear translocation, and thus, only the steady-state partitioning between nucleus and cytoplasm is observed. Nonetheless, delocalisation of IPO4 from the mitochondrial network and concomitant nuclear accumulation of H3.1-EGFP suggests a hand-off event between the importin and the nuclear histone chaperoning machinery.

Taken together, these findings suggest that a number of importin-β proteins could interact with cytosolic H3.1 and H4 and, in the absence of the core histone chaperoning machinery in the cytosol, supports the idea that importin-β family members play both a nuclear receptor and histone chaperoning function (Jakel *et al*, 2002).

## Quantification of chaperone-bound H3 and H4 reveals a pool of monomeric H3

Our RAPID-release approach demonstrated that *artificially* tethered H3.1 and H4 are predominantly monomeric in the cytosol and can be rapidly imported into the nucleus, suggesting that a monomeric pool of nuclear H3.1 and H4 may exist within the cell. To corroborate our findings from the RAPID-release approach, we wondered whether this pool could be isolated biochemically. To address this, we performed one-step pulldown analysis of EGFP-tagged chaperoning components (sNASP, ASF1b and HAT1) combined with quantitative analysis of histone stoichiometry, as a skew in the stoichiometry of H3 to H4 would be indicative of a monomers being present. Fusion proteins were nuclear in HeLa generated stable cell lines, confirming that these proteins are nuclear and not cytoplasmic in any quantity in interphase cells (Fig 6A). Pulldowns were performed on whole cell soluble extracts, with H3 and H4 clearly discernible in sNASP and ASF1b samples when stained with Coomassie (Fig 6B). Much less soluble H3 and H4 were associated with HAT1, which may represent a more transient interaction with histones.

Analysis of recombinantly purified histones revealed a wide linear relationship between Coomassie staining and protein concentration, with the relative molar ratios of H3 to H4 remaining stable up to at least 400 ng (Fig EV5A–D). When this linear window was

---

**Figure 5. Importin-β family as stable interactors of monomeric H3.1 and H4.**

A  Colocalisation between importin-β proteins and tethered histone H4. Representative images for each importin are shown. Scale bar represents 10 μm.

B  Pearson's correlation analysis between tethered H3.1/H4-EGFP and importin-β proteins. *P*-values of < 0.001 from a Wilcoxon rank sum test versus the mCherry alone control are indicated by brackets.

C  Phylogenetic representation of the importin-β family from humans adapted from O'Reilly *et al* (2011). For further details, see the main text. Members found to bind to either H3.1 or H4 in this study are coloured red. An asterisk indicates structural information regarding interaction with H3 (see Fig EV4).

D  FRAP analysis of mCherry-IPO4 and mCherry-RbAp46-NES bound to tethered EGFP-H4. Scale bars represent 10 μm.

E  FRAP analysis of mCherry-IPO4 and mCherry-sNASP-ΔNLS bound to tethered EGFP-H3. Scale bars represent 10 μm.

F  Time course of H3.1-EGFP release from its mitochondrial tether after co-transfection with mCherry-IPO4.

G  Time course in (D) shown as a profile that bisects the nucleus and the mitochondrial network. The location of the profile is shown as a white line in the lower right panel of (D). Profile correlation scores for each time point are shown.

H  Quantification of nuclear enrichment over time for H3.1-EGFP and mCherry-IPO4 from five individual cells. H3.1-EGFP values are green traces, whereas mCherry-IPO4 traces are red traces. Values for the cell shown in (F) are highlighted by circles.

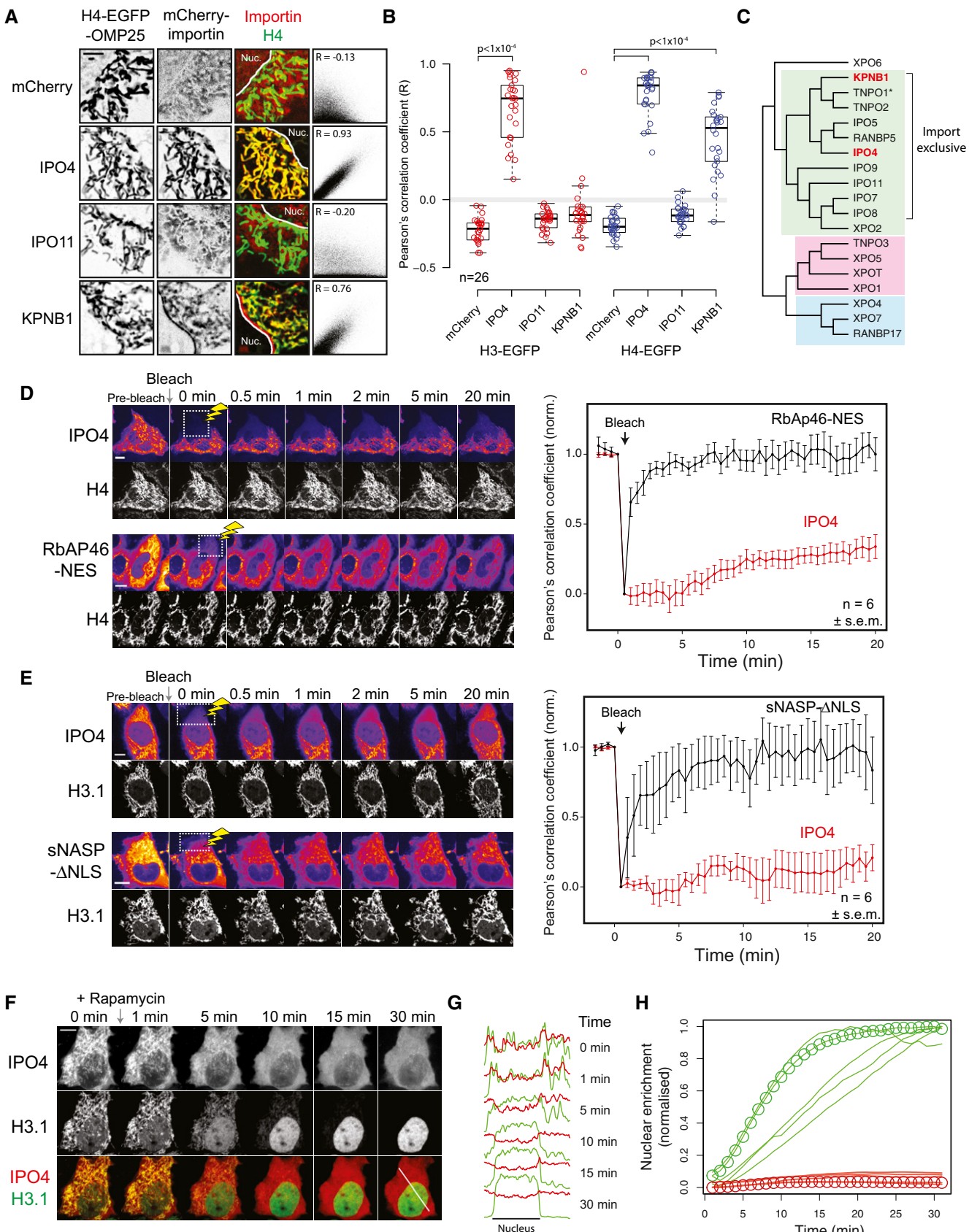

**Figure 5.**

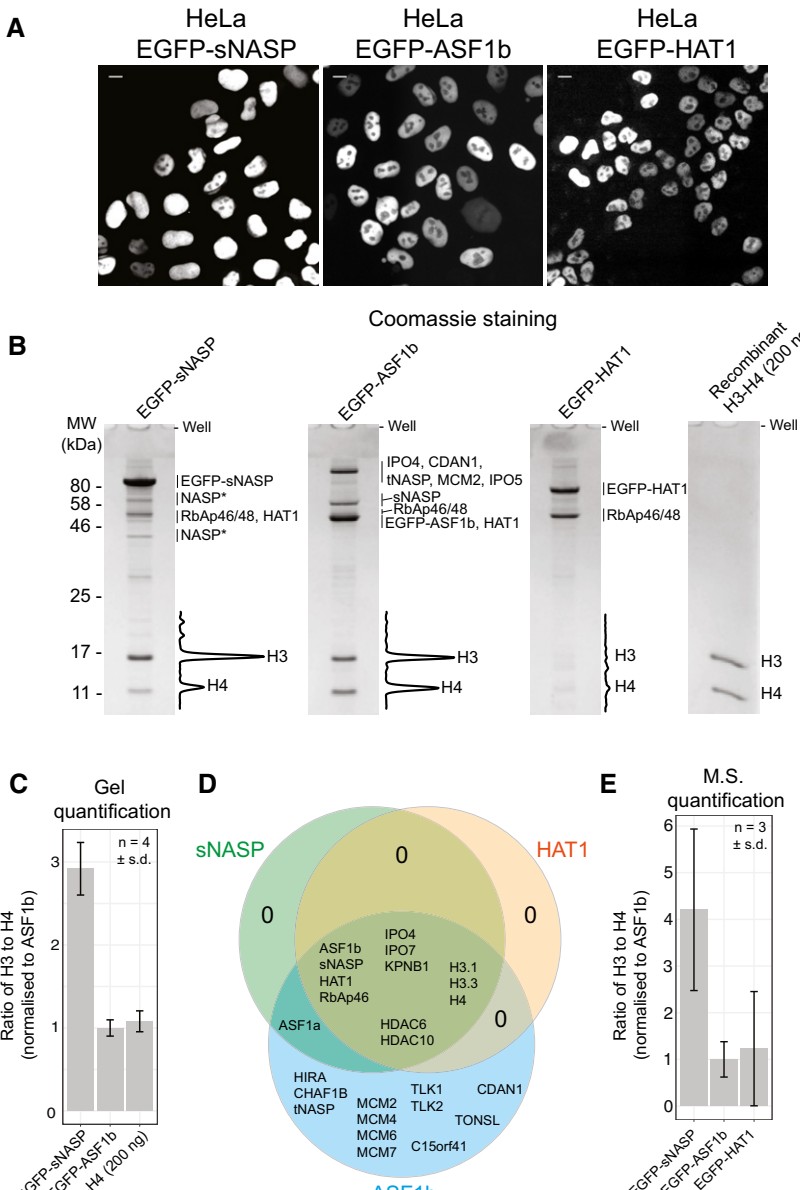

**Figure 6.   Quantification of histone stoichiometries bound to sNASP, ASF1b and HAT1.**

A   HeLa cell lines stably expressing EGFP-sNASP, EGFP-ASF1b and EGFP-HAT1. Scale bars represent 10 μm.

B   Histone chaperone immunoprecipitations separated by 15% SDS–PAGE and stained with Coomassie. Major bands that could be confidently identified by mass spectrometry are indicated. Due to the high percentage of acrylamide used, many of the higher molecular weight bands are not resolved. Asterisks represent partial degradation. Background-corrected densitometry profiles are shown for the portions of the gels covering H3 and H4. Gel filtration purified recombinant H3-H4 dimer is shown on the right.

C   Quantification of the relative molar ratios of H3 to H4 from Coomassie-stained gels. Four biological replicates were made. Values are normalised to ASF1b. HAT1 is omitted due to histone levels being difficult to discriminate from background staining.

D   Venn diagram displaying the overlap of interactors associating with the three immunoprecipitated proteins.

E   Quantification of the relative molar ratios of H3 to H4 from mass spectrometry analysis of three of the biological replicates using label-free quantification (LFQ). Values are normalised to ASF1b.

Source data are available online for this figure.

used to quantify the relative molar ratio of H3 to H4, we found that, whilst ASF1b corresponded well with a 1:1 ratio as compared to recombinant H3-H4 dimer, a significant excess of H3 came down with sNASP, corresponding to a relative molar ratio of 3:1, or three times as much H3 relative to H4 (Fig 6C). Histones associated with HAT1 were not sufficiently above background to allow for accurate quantification by this method, with only a faint H4 band visible, and no discernible band for H3 (Fig 6C).

Analysis of three of the replicates by mass spectrometry revealed a well-documented co-chaperoning network (Hammond *et al*, 2017), with all three proteins sharing interactions with each other and with a select group of other factors including the second HAT1-complex component RbAp46, importin-β proteins IPO4, IPO7 and KPNB1, histones H3.1, H3.3 and H4, in addition to the histone deacetylases HDAC6 and 10 (Fig 6D). As the soluble extract is a mix of both cytosolic and nuclear components, it is likely that the importin-β proteins, which are predominantly cytosolic under steady-state conditions, represent a re-association with complexes post-lysis, either directly to histone tails or via nuclear localisation sequences within the histones chaperones. Interestingly, we did not identify any accessory factors strongly associated with sNASP or HAT1, but identified numerous factors unique to ASF1b, including the histone chaperones HIRA, CHAF1B and tNASP, components of the MCM helicase, the kinases TLK1 and 2, CDAN1 and TONSL, all of which have been previously identified (Sillje & Nigg, 2001; Mello *et al*, 2002; Tagami *et al*, 2004; Groth *et al*, 2007; Duro *et al*, 2010; Ask *et al*, 2012; Klimovskaia *et al*, 2014). In addition, we found a novel, undocumented interactor, C15orf41, mutations of which have been implicated in congenital dyserythropoietic anaemia (Babbs *et al*, 2013), similar to CDAN1. This analysis highlights the core networking role of ASF1 in the histone chaperoning pathway, with NASP and the HAT1 complex most likely having dedicated roles in histone chaperoning and acetylation.

The mass spectrometry analysis also allowed us to quantify the ratios of H3 to H4 using label-free quantification (LFQ), which, when normalised to ASF1b, revealed a molar ratio corresponding to 4:1, or four times as much H3 relative to H4 (Fig 6E), comparable to the values obtained from Coomassie staining (compare Fig 6C with E). Whilst both H4 and H3 peptides were identified in the HAT1 pull-down, the coverage and peptide intensities for H3 were much lower and less reproducible than for sNASP and ASF1b, with variations close to the baseline detection resulting in significant error levels. Thus, whilst H4 appeared in excess of H3 from Coomassie staining, an accurate stoichiometry using LFQ could not be obtained (Fig 6E).

In summary, whilst dynamic information about the histone chaperoning pathway is difficult to obtain by biochemical means, our quantitative analysis of chaperone components under steady-state conditions reveals that sNASP interacts with a significant pool of monomeric H3 within the nucleus, lending further support for a biological role of histones in their monomeric conformation.

# Discussion

### Design principles for the RAPID-release approach

To observe dynamics of newly synthesised histones in living cells, we needed a system that (i) enabled labelling of histones in the cytosol, (ii) allowed a controlled and expeditious release of the tethered cargo, and (iii) was well tolerated by the cell. As histone import is a fast process, we rationalised that sequestration of histones in the cytoplasm soon after synthesis would be necessary to allow sufficient time for labelling of the histones (i.e. the folding and maturation of an EGFP tag), and to build up a significant reservoir of labelled protein to observe a pulse-chase event. We achieved this by utilising a mitochondrial tail-anchoring peptide (Horie *et al*, 2002),

which, although worked well, necessitated peptide bond cleavage to release the tethered histones. We also reasoned that gating of the cleavage would be necessary to control the pulse-chase and that this would most likely involve a small molecule-mediated switch. We considered a number of small molecule-activated protease systems such as the allosteric activation of the cysteine protease domain (CPD) of the *Vibrio cholerae* RTX toxin (Lupardus *et al*, 2008) and the activation of a split TEV protease through rapamycin-controlled peptide complementation (Wehr *et al*, 2006; Gray *et al*, 2010). However, in the case of the RTX toxin, the small molecule ligand (inositol hexakisphosphate) is present in the cytosol of mammalian cells, and in the case of the split TEV protease, folding after complementation requires tens of minutes to hours for full activation (Kerppola, 2006; Wehr *et al*, 2006; Gray *et al*, 2010). Instead, we pursued a proximity sensor approach in which an auto-inhibited TVMV protease is both recruited and activated through the addition of rapamycin (Stein & Alexandrov, 2014, 2015). The auto-inhibited TVMV protease was well tolerated by the cell, effectively inactive in its soluble form, but highly active when recruited to its substrate, serving as an excellent switch for the release of the tethered histones.

In comparison with self-labelling tags, such as the SNAP-/HALO-tags, RAPID-release is compatible with any protein or peptide tag fusion. This may find uses in expanding the functionality of pulse-chase proteins through the addition of novel enzymatic fusions (for example, in proximity ligation; Kim & Roux, 2016) or may prove beneficial in enabling a pulse-chase experiment to be performed with smaller peptide tags, which often show less interference than larger globular tags such as GFP. In addition, the kinetics of RAPID-release is typically an order of magnitude faster than what is achieved with SNAP-tag-derived methodologies (Jansen *et al*, 2007; Clement *et al*, 2016). However, even with this increase in pulse kinetics, the rate of histone nuclear import was still in excess of the kinetics of histone release. Thus, to effectively model rapid cellular processes, such as the import dynamics of histones, improvements with regard to cleavage kinetics will need to be made. In this regard, the downside to a viral protease's stringent specificity is its lower catalytic turnover, although a reported turnover rate of 11 s for TVMV suggests a significant increase in release kinetics is still achievable (Hwang *et al*, 2000; Nallamsetty *et al*, 2004; Sun *et al*, 2010). Despite these drawbacks, the ability to cytosolically tether H3.1 and H4 allowed us to investigate the nucleo-cytoplasmic divide at a level that has previously not been possible, bypassing the requirement for biochemical analysis, and enabling the observation of core histone incorporation into chromatin at actively replicating domains in living cells.

### Reassessing the nucleo-cytoplasmic divide within the histone chaperoning pathway

A number of interesting findings came from the ability to tether histones in the cytosol, most notably the stable monomeric nature of the tethered histones and the absence of previously identified cytosolic histone chaperones. The explanation we favour to explain this is that the previously identified histone chaperones are nuclear in living cells, but rapidly leak from the nucleus upon fractionation, giving the impression they are cytosolic when analysed by biochemical methods. A previous investigation into the nuclear import of the linker histone H1 put forward the idea that members of the importin-β family may have a dual role in both chaperoning and

nuclear localisation of basic proteins as they traverse to the nucleus (Jakel *et al*, 2002). We probed a number of importin-β proteins and found IPO4 and KPNB1 associated with cytosolically tethered H3.1 and/or H4, but do not accumulate in the nucleus upon histone release, suggesting a rapid hand-off event to the nuclear chaperoning machinery. Interestingly, current evidence regarding the binding sites of importin-β proteins suggests they interact with regions in the histone tails (Baake *et al*, 2001; Mosammaparast *et al*, 2002; Blackwell *et al*, 2007; Soniat & Chook, 2016; Soniat *et al*, 2016), with a recent crystal structure identifying the H3 epitope that binds to TNPO1 (Soniat & Chook, 2016) and H3 and H4 epitopes that bind to the yeast homolog Kap123 (An *et al*, 2017). Dumping of monomeric histones in the nucleus is unlikely; therefore, the molecular mechanism of how to transfer between importins and chaperone, and how this is coordinated with Ran-GTP binding, may be of future interest.

Interestingly, our quantitative analysis of H3-H4 stoichiometries bound to nuclear histone chaperones revealed that a significant excess of H3 over H4 associates with the chaperone sNASP. A previous study demonstrated that sNASP can bind both monomeric H3 and an H3-H4 dimer in complex with ASF1 *in vitro* (Bowman *et al*, 2017). Interaction with the monomeric H3 is mediated in part through NASP's TPR domain binding to the C-terminus of H3 (Bowman *et al*, 2016). As importins interact with the N-terminal region of histones, the C-terminal region of H3 would be free to hand-off to NASP. In this respect, sNASP may act as a nuclear *receptor* for newly synthesised monomeric H3, presenting an interesting direction for future investigations. Similarly, whether a nuclear receptor for monomeric H4 exists has also to be further investigated; however, a number of soluble nuclear proteins have been shown to bind to H4 motifs in the absence of H3, including HAT1 (H4 residues 5–18) (Wu *et al*, 2012), RbAp46 (Murzina *et al*, 2008; Song *et al*, 2008) and TONSL (H4 residues 12–23) (Saredi *et al*, 2016). Whilst we could see a qualitative excess of H4 over H3 associated with HAT1 after SDS–PAGE separation, further studies will need to address this satisfactorily.

A corollary of the HAT1 complex residing in the nucleus is that newly synthesised H4 would not be acetylated on K5 and K12 until it is imported. Although studies have reported the role of lysine to glutamine mutations (as acetyl-lysine mimics) in aiding nuclear import in *P. polycephalum* (Ejlassi-Lassallette *et al*, 2011), a recent crystal structure has revealed the importance of unacetylated H4 tail peptides in mediating interaction with the *K. lactis* importin protein Kap123 (An *et al*, 2017). In addition, whilst crystal structures have revealed the importance of histone tail regions in binding a number of importin-β proteins (Soniat & Chook, 2016; An *et al*, 2017), our truncation analysis would suggest that the junction with the core domain is of more importance in efficient nuclear delivery. This is supported by a recent study demonstrating that tail swaps result in lower rates of nuclear import (Ejlassi *et al*, 2017). Whether importin-β proteins make more extensive interactions with the core domain of histones, and thereby acting as chaperones in their own right, will require further investigation.

In addition to a potential role in the thermodynamic assembly of prenucleosomal subunits, a provision for sequestration of monomeric histones could aid the cell in coping with non-stoichiometric increases in soluble histones during periods of replication stress, where DNA synthesis is suddenly halted, or in other instances which produce an imbalance in the stoichiometries of core histones.

Members of canonical protein folding pathways, such as HSP90 and HSC70, may also aid in such processes (Campos *et al*, 2010) or could potentially be involved in excess histone degradation (Cook *et al*, 2011).

A caveat of our reassessed model is that we cannot explicitly exclude the possibility that H3.1 and H4 rapidly fold with their endogenous histone counterparts in the cytosol after release, but before nuclear import can occur. We think this unlikely, however, as cells outside of S-phase, which undergo limited histone synthesis, followed comparable import kinetics to those inside of S-phase. Secondly, release of a large quantity of tagged histones is likely to transiently quench the soluble pool of the endogenous counterpart, yet we found nuclear import occurred rapidly, without a soluble accumulation of histones in the cytoplasm. Finally, we could detect *supra*-stoichiometric amounts of H3 associated with NASP, suggesting that a soluble pool of monomeric H3 exists in the nucleus and thus lending further credence to the idea of monomeric histone import.

In summary, our findings suggest a revised model (depicted in Fig 7) in which H3 and H4 are imported into the nucleus as monomers bound tightly to importin-β proteins, wherein they are transferred to the dedicated histone chaperoning network and fold to form an H3-H4 heterodimer for incorporation into chromatin.

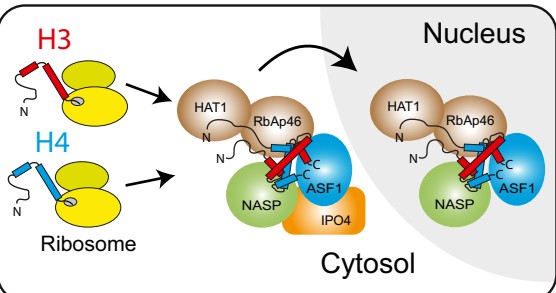

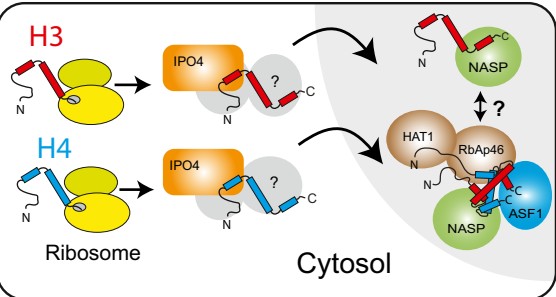

**Figure 7.  Contrasting models for histone chaperoning during nuclear import of H3.1 and H4.**

In the dimer import model, H3.1 and H4 form a heterodimer soon after synthesis in the cytosol. This heterodimer is bound by the chaperones NASP, ASF1, RbAp46 and HAT1, which further associate with IPO4 before translocation to the nucleus. In the monomer import model, H3 and H4 monomers are bound by importin-β proteins before translocation to the nucleus, wherein they form a heterodimer and associate with the core histone chaperoning machinery. Additional cytosolic factors may also transiently interact with the monomeric histones.

# Materials and Methods

### Cloning and vector assembly

EGFP$^{TVMVx2}$-FRB-OMP25, H4-EGFP$^{TVMVx2}$-FRB-OMP25, H3.1-EGFP$^{TVMVx2}$-FRB-OMP25, H3.1-mCherry$^{TVMVx2}$-FRB-OMP25, EGFP-FRB-OMP25, mCherry-FKBP12-TVMV-AI, mCherry-FKBP12-TVMV and TagBFP-FKBP12-TVMV-AI, H3(FLY>AA)-EGFP$^{TVMVx2}$-FRB-OMP25 and H3.1 and H4 truncations were constructed through a mixture of PCR cloning, annealed oligo ligation, gBLOCK synthesis (IDT) and Gibson assembly (Gibson *et al*, 2009) using the mammalian expression vectors pEGFP-C1/C3, pmCherry-C1 and pTagBFP-C1 and piFP2.0-C1 (Yu *et al*, 2014). FKPB12-TVMV-AI and $^{TVMVx2}$-FRB cassettes were based on published sequences (Stein & Alexandrov, 2014).

Open reading frames from IPO4, KPNB1 and IPO11 were amplified from U2OS cDNA and inserted into the HindIII site of pmCherry-C1 using Gibson assembly. EGFP fusions of RbAp46 and CAF1p60 were created by amplifying open reading frames from U2OS cDNA and ligating them into the HindIII-KpnI sites of pEGFP-C1. HAT1 was first cloned into the protein expression vector pETMCN6His using the NdeI-BamHI sites and then subcloned into pEGFP-C1 using the restriction sites KpnI and BamHI. This transferred the TEV cleavage site from the pETMCN6His vector, extending the linker between HAT1 and EGFP. EGFP-sNASP and EGFP-ASF1a/b were cloned previously (Bowman *et al*, 2016, 2017). EGFP-sNASP/-HAT1/-ASF1b fusions were then transferred to the multiple cloning site of the pIRESpuro2 vector (Clonetech) using NheI-BamHI sites. mCherry-sNASP-ΔNLS was created by subcloning from a published EGFP-fusion vector (Bowman *et al*, 2016) into the pmCherry-C1 vector using KpnI and BamHI sites. pmCherry-NES-RbAp46/-HAT1/-ASF1A was created by first inserting a NES sequence into the multiple cloning site of pmCherry-C1 using annealed oligos, followed by subcloning of the chaperones from pEGFP-C1 vectors as described above.

The vector encoding PCNA-V$_H$H-TagRFP chromobody was purchased from Chromotek Gmbh, Munich. Lamin A/C was amplified from a cloned gene and inserted into the vector pEGFP-C1 using Gibson assembly. For a detailed list of plasmid constructs and reagent, please refer to the Expanded View.

### Tissue culture and transfection

HeLa cells originally obtained from ATCC (HeLa, ATCC® CCL-2™) were expanded and frozen as aliquots in liquid nitrogen as a source stock. Cells were cultured in DMEM (produced in-house) supplemented with 10% heat-inactivated foetal bovine serum (Sigma), 4 mM glutamine and 50 μg/ml penicillin/streptomycin at 37°C in a humidified incubator with 5% $CO_2$. Cells were passaged with 0.25% trypsin-EDTA, centrifuged (180 *g* for 3 min) and then resuspended in complete media. Cell counts were performed using a haemocytometer.

Transfections were performed following the Fugene HD manufacturer's instructions (Promega). The day before transfection, HeLa cells were passaged, counted and plated at an appropriate density ($2.5 \times 10^5$ or $2.5 \times 10^4$ cells per well for a 6-well plate and an 8-well μ-slides (ibidi), respectively). Transfection mixtures contained 1 μg plasmid DNA in 50 μl DMEM and then vortexing with 3 μl Fugene HD transfection reagent. The transfection mixture was incubated at room temperature for 10 min prior to addition to cells. Cells were transfected with 15 or 50 μl of the transfection mixture per well for an 8-well μ-slide or 6-well plate, respectively. Cells were imaged or fixed 24 h post-transfection.

Stable cell lines of EGFP-tagged sNASP, ASF1b and HAT1 were created by transfecting HeLa cells with the corresponding pIRE-Spuro2 vector and isolating individual clones after 2 weeks of selection in 0.3 μg/ml puromycin. Clones were initially screened for nuclear fluorescence before immunoprecipitation and identification of the correct size band using SDS–PAGE.

### Biochemical fractionation of HeLa cells

Cell fractionation was performed following the REAP protocol (Suzuki *et al*, 2010). Concisely, HeLa cells cultured in a 100 mm culture dish were washed twice with PBS before being scraped off in 1 ml ice-cold PBS and collected in a 1.5 ml Eppendorf tube. The cells were pelleted by centrifugation for 10 s, the supernatant removed and then the cells resuspended in lysis buffer (0.1% NP-40 in PBS containing protease inhibitors). A whole cell lysate sample was removed before the lysate was briefly centrifuged for 10 s and the supernatant taken as the cytoplasmic fraction. The pellet was resuspended in lysis buffer, centrifuged for 10 s and the supernatant removed. The pellet which contained the nuclear fraction was resuspended in lysis buffer. The cell fractions were diluted in Laemmli sample buffer and boiled for 3 min. Samples of the cellular fractions were run on a 15% acrylamide gel. Proteins were transferred onto a nitrocellulose membrane and then incubated at 4°C overnight with primary antibodies diluted in 5% milk or 3% BSA (all 1:1,000 except anti-NASP 1:10,000). Membranes were washed with TBST and then incubated for 1 h at room temperature with the HRP-conjugated secondary antibody.

### Purification of immunogens

6His-sNASP and GST-ASF1A were expressed and purified as described previously (Bowman *et al*, 2016, 2017). The peptide immunogen against which the RbAp46, HAT1 and CHAF1B antibodies were raised (RbAp46 residues 1–19, and HAT1 residues 320–420, CHAF1B residues 441–552, respectively) were expressed as MBP fusions from a pMAL-CRI-derived vector in Rosetta 2 (DE3) cells (Novagen) and purified over Dextrin Sepharose (GE Healthcare) according to manufacturer's instructions. The MBP fusions were then used directly in the blocking experiments without cleavage of the affinity tag.

### Imaging

All images were captured using an UltraVIEW VoX Live Cell Imaging System (PerkinElmer). Live cell imaging involved culturing HeLa cells in 8-well μ-slides (ibidi) and replacing medium with 200 μl Leibovitz's L-15 medium (Thermo Fisher Scientific) supplemented with 4 mM L-glutamine and 50 μg/ml penicillin/streptomycin. For RAPID-release experiments, 50 μl Leibovitz's medium containing 1 μM rapamycin was added directly to the well resulting in a final concentration of 200 nM rapamycin.

For imaging nuclear leakage during cell lysis, cells were imaged 24 h post-transfection with mCherry-sNASP and EGFP-Lamin A/C.

Hoechst at a concentration of 500 ng/ml was added to the cells prior to imaging and incubated for 20 min. Culture medium was replaced with PBS immediately prior to imaging. An initial capture of 20 *z*-stacks penetrating 20 μm into the culture dish was acquired before the addition of NP-40 to 0.1% or replacement of PBS with $H_2O$. *Z*-stacks were then acquired every minute for up to 15 min. Orthogonal views were created in ImageJ using the Reslice function. For quantification, *Z*-stacks were flattened using a maximum intensity projection, and the fluorescence as a percentage of maximum for the whole time series was plotted as the mean of normalised values from a total of 15 cells (Fig 4E). The error bars represented the standard error of the mean (SEM).

For immunofluorescence, HeLa cells cultured on glass coverslips in 6-well plates were fixed with 4% paraformaldehyde for 10 min. Unreacted formaldehyde was quenched with 50 mM ammonium chloride for 10 min before the cells were permeabilised with 0.2% Triton X-100 in PBS for 15 min. Cells were blocked in 3% BSA in PBS for 1 h and subsequently incubated with the primary antibody diluted in 1% BSA in PBS overnight at 4°C. Excess antibody was removed with successive washes with 0.1% Tween-20 in PBS and then incubated with the appropriate secondary antibody for 1 h at room temperature. After removal of excess secondary antibody, cells were stained with 1 μg/ml Hoechst in PBS for 10 mins and then coverslips were mounted using Prolong Gold antifade reagent (Thermo Fisher Scientific). Antibody concentrations used were all 1:100 for primary antibodies except anti-NASP (1:5,000) and 1:500 for secondary antibodies. For antibody blocking experiments, recombinant immunogens for each antibody were expressed and purified from bacteria (see below) and added to antibodies at a 100-fold molar excess prior to immunostaining.

### Nuclear enrichment analysis

CellProfiler (http://cellprofiler.org/) was used to analyse images of HeLa cells immunostained for histone chaperones with and without immunogen blocking. Nuclei were segmented using Hoescht staining as a mask, with the 20 pixels surrounding the demarcated nuclei taken as the cytoplasmic region. The mean intensities of the nuclear and cytoplasmic regions were used to calculate the nuclear-cytoplasmic ratio.

### RAPID-release methodology and analysis

An initial image stack was taken prior to rapamycin addition, serving as time 0, after which cells were imaged every minute for up to 30 min. Partitioning and quantification of the images was carried out using ImageJ. Exponential, logarithmic and linear models were fitted to the data using the statistical analysis software R.

For RAPID-release of tethered H3.1-EGFP and soluble mCherry-IPO4, cells were co-transfection with H3.1-EGFP$^{TVMVx2}$-FRB-OMP25, mCherry-IPO4 and IFP2.0-FKBP12-TVMV-AI with release being performed as described above. Expression of IFP2.0-FKBP12-TVMV-AI was checked by addition of 20 mM biliverdin 60 min before imagining cells with a 640 nm laser, allowing H3.1 and IPO4 to be followed in the green and red channels, respectively. *Z*-stacks spanning the cell were flattened into a maximum pixel intensity image. The cytosol and nucleus were manually partitioned for each cell, and the nuclear enrichment over the cytosol was calculated for each time point. Values for individual cells were normalised between 1 and 0 and plotted on the same axes for comparison.

### Colocalisation at sites of active replication

Asynchronously growing cells were transiently co-transfected with H3.1/H4-EGFP$^{TVMVx2}$-FRB-OMP25, TagBFP-FKBP12-TVMV-AI and PCNA-V$_H$H-TagRFP using FugeneHD (Promega), as described above. 24 h after transfection, histones were released by the addition of rapamycin (200 nM) to the cell culture medium at 5, 2.5, 1 h and 30 min before washing with PBS and fixing in 4% PFA in PBS. Cells were imaged using confocal microscopy as described above, taking 10 z-sections that spanned the volume of the nucleus. Cells were manually screened for nuclei that displayed a peripheral pattern of PCNA staining, indicating mid-to-late S-phase. Due to the large amount of background space outside of the PCNA foci, single Z-slices that bisected the centre of the nucleus were processed to extract masks for the PCNA and histone-EGFP channels using the ImageJ plugin "FindFoci" (Herbert *et al*, 2014). The masks were then used in CDA analysis (Ramirez *et al*, 2010) using the ImageJ CDA plugin to calculate the Pearson's correlation coefficient (R) for each cell (http://www.susse x.ac.uk/gdsc/intranet/microscopy/imagej). Boxplots of the R-values were created using the program R, and the Wilcoxon rank sum test was carried out to determine the significance of the difference in colocalisation over time. *P*-values of < 0.001 are displayed in Fig 3D.

### Colocalisation in the cytosol at the mitochondrial network (mitochondrial 2-hybrid)

Cells were either imaged 24 h post-transfection or fixed 24 h post-transfection and stained with the corresponding antibody. A single Z-plane through the cytosol was taken for colocalisation analysis. Cytosolic regions encompassing the mitochondrial network were manually partitioned and a 20-pixel rolling ball background correction applied (ImageJ) (Fig EV3A). Pearson's correlation coefficients (R) were calculated using the GDSC Colocalisation Threshold plugin (http://www.sussex.ac.uk/gdsc/intranet/microscopy/imagej/coloca lisation). Due to the extensive spread of the mitochondrial network through the cytosol, image masks were not employed. R$_{total}$ values, without thresholding, were used. Boxplots of the R values were created using the program R, and the Wilcoxon rank sum test was carried out to determine the significance of colocalisation.

### Fluorescence recovery after photobleaching (FRAP)

H3.1-/H4-EGFP$^{TVMVx2}$-FRB-OMP25 and mCherry fused IPO4, RbAp46-NES or sNASP-ΔNLS constructs were co-transfected into HeLa cells seeded onto glass bottomed 8-well μ-slides (ibidi). FRAP experiments were performed using a spinning disc confocal microscope (UltraVIEW VoX Live Cell Imaging System, PerkinElmer) equipped with a PhotoKinesis FRAP module and a 37°C environmental chamber in Leibovitz's L-15 medium (Thermo Fisher) supplemented with 4 mM glutamine and 10% foetal calf serum (Sigma). Cells with evenly spread mitochondrial networks were chosen for analysis. Four pre-bleaching images of a single z-slice were taken before rectangular regions approximating 25–50% of the cell area were bleached using 30 iterations of the 561 nm laser (the number of iterations determined by scouting experiments to sufficiently bleach

without cause cytotoxicity). Images were taken directly after bleaching, and every 30 s subsequently, up to 20 min. The Pearson's correlation coefficients between the EGFP (tethered histone) and the mCherry (IPO4, Rbap46-NES or sNASP-ΔNLS) channels were calculated for the bleached area at each time point using the ImageJ Stack Colocalisation Analyser plugin (http://www.sussex.ac.uk/gdsc/intranet/microscopy/imagej/colocalisation). Cells in which mitochondria were observed to migrate into the bleached area during recovery were discarded. Values were normalised, taking the last pre-bleach image as 1 and the first post-bleach image as 0. Error bars represent the s.e.m. of six individual experiments.

### Immunoprecipitation of EGFP fusions

For each biological replicate, three 15-cm dishes of cells were grown to 70% confluency in DMEM (produced in-house) supplemented with 10% heat-inactivated foetal bovine serum (Sigma), 4 mM glutamine, 50 μg/ml penicillin/streptomycin and 0.3 μg/ml puromycin at 37°C in a humidified incubator with 5% $CO_2$. Cells were harvested with 0.25% trypsin-EDTA, washed twice in ice-cold PBS, made up to 10 μg/ml with aprotinin (to inhibit residual trypsin activity), pelleted and lysed by resuspension in PBS + 0.1% NP-40 + protease inhibitors. Lysis continued for 10 min on ice, before clearing of the lysate by centrifugation and binding of the soluble fraction to 20 μl of GFP-Trap® (ChromoTek) agarose beads. Binding was allowed to proceed for 30 min at 4°C with continual nutation. Beads were washed once in 20 mM Tris–HCl pH 7.5, 400 mM NaCl and 0.1% NP-40 (high salt buffer), followed by four washes in PBS + 0.1% NP-40, omitting the NP-40 from the last wash. Beads were then boiled in 20 μl of 2× Laemmli buffer with the entire pulldown loaded into a single well of a 15% polyacrylamide gel. Gels were stained with Coomassie (InstantBlue™, expedeon).

### Sample preparation and nanoLC-ESI-MS/MS analysis

Gels from immunoprecipitation experiments were cut into fragments and in-gel digested (Shevchenko *et al*, 2006). The peptides were analysed with two columns, an Acclaim PepMap μ-precolumn cartridge 300 μm i.d. × 5 mm length, 5 μm particle size, 100 Å pore size and an Acclaim PepMap RSLC 75 μm i.d. × 50 cm, 2 μm, 100 Å (Thermo Scientific). The columns were installed on an Ultimate 3000 RSLCnano system (Dionex) at 40°C. Mobile phase buffer A was composed of 0.1% formic acid, and mobile phase B was composed of acetonitrile containing 0.1% formic acid. Samples were loaded onto the μ-precolumn equilibrated in 2% aqueous acetonitrile containing 0.1% trifluoroacetic acid for 5 min at 10 μl/min after which peptides were eluted onto the analytical column at 250 nl/min by increasing the mobile phase B concentration from 8% B to 35% over 38 min, followed by a 2 min wash at 80% B and a 15 min re-equilibration at 4% B.

Eluting peptides were converted to gas-phase ions by means of electrospray ionisation and analysed on a Thermo Orbitrap Fusion (Thermo Scientific). Survey scans of peptide precursors from 375 to 1,575 *m/z* were performed at 120K resolution (at 200 *m/z*) with a $5 \times 10^5$ ion count target. The maximum injection time was set to 200 ms. Tandem MS was performed by isolation at 1.2 Th using the quadrupole, HCD fragmentation with normalised collision energy of 33 and rapid scan MS analysis in the ion trap. The $MS^2$ ion count

target was set to $8 \times 10^3$, and maximum injection time was 200 ms. Precursors with charge state 2–6 were selected and sampled for $MS^2$. The dynamic exclusion duration was set to 30 s with a 10 ppm tolerance around the selected precursor and its isotopes. Monoisotopic precursor selection was turned on, and instrument was run in top speed mode. The mass spectrometry proteomics data have been deposited to the ProteomeXchange Consortium via the PRIDE (Vizcaino *et al*, 2016) partner repository with the dataset identifier PXD009915.

### M.S. data analysis

The raw data were searched using MaxQuant software version 1.6.0.16 against UniProtKB Human database (UP000005640, 71,785 entries, release March 2017) and the common contaminant database from MaxQuant (Tyanova *et al*, 2016). Peptides were generated from a tryptic digestion with up to two missed cleavages, carbamidomethylation of cysteines as fixed modifications, protein N-terminal acetylation and methionine oxidations as variable modifications. Precursor mass tolerance was 10 ppm, and product ions were searched at 0.8 Da tolerances. Scaffold (TM, version 4.6.2, Proteome Software Inc.) was used to validate MS/MS-based peptide and protein identifications. Peptide identifications were accepted if they could be established at > 80.0% probability by the Scaffold Local FDR algorithm. Protein identifications were accepted if they could be established at > 90.0% probability and contained at least two identified peptides. Proteins that contained similar peptides and could not be differentiated based on MS/MS analysis alone were grouped to satisfy the principles of parsimony. Proteins sharing significant peptide evidence were grouped into clusters.

**Expanded View** for this article is available online.

### Acknowledgements

We would like to acknowledge the Warwick Proteomics RTP for mass spectrometry analysis; Robert Cross, Andrew McAinsh and Anne Straub for access to microscopy resources; Erick M. Ratamero for assistance in data plotting; Dejana Mokranjac for advice on tail-anchoring proteins; Robert Cross for the monoclonal tubulin antibody; and Andreas Ladurner for the polyclonal NASP antibody. This work was supported by the Wellcome-Warwick Quantitative Biomedicine Program (Institutional Strategic Support Fund: 105627/Z/14/Z) and a Sir Henry Dale Fellowship to A. J. B. (208801/Z/17/Z).

### Author contributions

AJB conceived of the study. AJB and MJA-S jointly carried out the experiments, analysed the data and wrote the manuscript. JRH-F contributed to sample preparation and analysis of mass spectrometry experiments.

### Conflict of interest

The authors declare that they have no conflict of interest.

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
