## [Review Process File · The EMBO Journal]

Evidence for the nuclear import of histones H3.1 & H4 as monomers

Michael James Apta-Smith, Juan Ramon Hernandez-Fernaud & Andrew James Bowman.

Review timeline:	Submission date:	24 th November 2017
	Editorial Decision:	17 th January 2018
	Revision received:	7 th June 2018
	Editorial Decision:	18 th July 2018
	Revision received:	20 th July 2018
	Accepted:	25 th July 2018

Editor: Anne Nielsen

Transaction Report:

1st Editorial Decision

17th January 2018

Thank you for submitting your manuscript for consideration by the EMBO Journal and for suggesting experiments that could be included in a revised version to address the referee concerns (following my previous email).

I have now looked at your response in light of the referee reports and discussed the matter with my colleagues in the editorial team. In addition, I have taken the liberty to also discuss your study with my colleague Andrea Leibfried, the executive editor of the new open-access journal Life Science Alliance. Life Science Alliance is launched as a partnership between EMBO Press, Rockefeller Press, and Cold Spring Harbor Laboratory Press, and publishes work that is of high value to the respective communities across all areas in the life sciences.

The outcome of these discussions is that we would like to give you the chance to submit a revised version of the study to The EMBO Journal, in which you address the referee concerns along the lines suggested in your email to me. However, at the same time I realise that the outcome of the chaperone IP experiment is rather open at this stage and that the referees may still be concerned about the conclusiveness for cytoplasmic histone stoichiometry. This is why I discussed your study with Andrea Leibfried at Life Science Alliance: she has offered to publish the study with very minor revision (the controls requested by referee #1 and the one for an unrelated OMM-tethered protein requested by referee #3) as well as text changes to highlight that the system is artificial. Furthermore, this offer would still be valid after revision in case the revision experiments turn out not to work or if the referees are not convinced about their conclusiveness.

At this point, I will therefore leave it up to you to decide if you want to pursue publication in The EMBO Journal by performing the experiments outlined in your response - with the caveat that the referees may still have some hesitations at a later stage - or transfer your manuscript to Life Science Alliance for rapid publication there. I realise that as a new journal, Life Science Alliance doesn't have much content to compare with at this stage but Andrea Leibfried would be happy to discuss the journal in more detail and explain where we see it scope-wise in the future.

In conclusion, I would like to invite you to submit a revised version of the manuscript, addressing the comments of all three reviewers. I should add that it is EMBO Journal policy to allow only a single round of revision, and acceptance of your manuscript will therefore depend on the completeness of your responses in this revised version.

REFEREE REPORTS.

Referee #1:

A method is employed to force nascent histones into the cytoplasm by tethering them to the outer membrane of mitochondria. Then, by an ingenious system that was previously developed, histones can be released from the outer mitochondrial membrane using a rapamycin inducible dimerization and protease activation system. This results in rapid nuclear import of the histones. The authors show that the system works and results in rapid translocation of histones into the nucleus upon Rapamycin induced cleavage. They go on to show that the degree of histone import correlates with the degree of Rapamycin dependent cleavage and that released histone are incorporated into chromatin in S phase.

The authors utilize this system to determine to what extent the known histone chaperones NASP, ASF1, HAT1 and RbAp46 associate with histone H3 and H4 while kept in the cytoplasm. They conclude that no such interaction takes place in the cytoplasm but only in the nucleus. They argue that prior data suggesting that such a prenucleosomal complex does exist in the cytoplasm is due to biochemical fractionation artifacts. When fractionating cells, NASP, RbAp46, HAT1, ASF1A and CAFp60 can all be found in the cytosolic pool but they authors argue that this is likely a soluble nuclear pool that is released into the cytoplasmic fraction. The experiments in figure 1 support this claim. Contrary to the histone chaperones, they find the nuclear receptors IPO4, IPO11 and KPNB1 to be associated with histones that are artificially tethered in the cytoplasm. Therefore, contrary to the currently literature, duly cited in this manuscript, the authors suggest that histone H3 and H4 do not form a heterodimer in the cytoplasm and do not interact with the canonical chaperones until they reach the nucleus.

The paper presents a series of well executed experiments and controls. The RAPID system, although artificial, allows the authors to arrest the very fast dynamics of nascent histone transfer to the nucleus, not possible with other techniques. The idea that histones complex with their cognate chaperones only in the nucleus is novel and relevant for the field. In my view this work should be published provided a few key issues are resolved, outlined below.

There is the potential caveat that there is a small fraction of histone chaperones present in the cytoplasm that goes beyond detection. The authors provided two sets of experiments indicating that this is unlikely:

1) To determine whether histone chaperones can, in principle, interact with cytoplasmic histones, mutant chaperones lacking an NLS or gaining an NES are expressed. These do interact with cytoplasmically tethered histones. To test whether this interaction is relevant, the authors use FRAP experiments in Figure 5D-E as a measure of the strength of the interaction which they show is weak compared to e.g. important IPO4. In principle, this is a good approach to demonstrate that histone chaperones are unlikely to interact with histones in the cytoplasm. However, in these experiments RbAp46 with a NES or sNASP lacking an NLS is broadly localized to the cytoplasm and it is not clear whether the FRAPed pool represents the pool bound to H4 or H3.1, respectively. A control should be included to show that the localization of these mutant chaperones is histone dependent by expressing them in cells lacking the tethered histones.

2) Another critical experiment that the authors provide as evidence that the prenucleosomal complexes in the cytoplasm are different from the ones in the nucleus is shown in figure 6A-E. The authors claim that H3 and H4 are monomeric in the cytoplasm by using a mutant form of H3 that is predicted to not interact with H4. It is clear that this mutant does not incorporate into chromatin yet is still translocated to the nucleus. However, whether it can still interact with H4 is not tested. This should be demonstrated.

Further, the inability of H3 to co-recruit H4 to the mitochondria (and visa versa) (figures 4C and EV3A) is taken as evidence that H3 and H4 are monomeric in the cytoplasm. However, it is also possible that the non-overexpressed partner is simply too low in abundance in the cytoplasm to detect. In the case of expressing mutant cytoplasmic histone chaperones, do the authors see co-recruitment of H3 to H4 or visa versa? This would control for the ability to detect these histones, if present.

Manuscript structure:

The presentation of the discrepancy of nuclear fractionation versus imaging of histone chaperons is confusing. In the abstract this is presented as one of the final observations while in the text the authors start by outlining this discrepancy as a starting point of their work. It would help if this would be streamlined. The first paragraph is entitled: "Mislocalization of histone chaperones during fractionation necessitates an in vivo pulse labelling system" It is not clear from the paragraph that the mislocalization during fractionation necessitates this system. The authors simply explain why chaperones "appear" cytosolic while being in fact nuclear but they should spell out more clearly how their RAPID system can resolve this issue.

Referee #2:

In this manuscript, Apta-Smith and Bowman test the possibility that histone H3.1 and H4 can be transferred into the nucleus as monomers. To this end they have developed an artificial tethering system of overexpressed monomeric tagged histones to mitochondria. Upon rapamycin treatment, tagged histones are rereleased, transferred to the nucleus and incorporated behind replication forks. The key findings are: 1) Endogenous member of the H3.1-H4 dimer (Endogenous H4 for H3.1-tag, and endogenous H3.1 for H4-tag) cannot be detected, suggesting that H3.1 and H4 can exist as monomers in the cytoplasm (Figure 4C). 2) Histones can be imported in the nucleus as monomer but not assembled onto newly replicated DNA (Figure 6C-E). I do not recommend this manuscript for revision for several reasons, listed below:

Main concerns

- The first finding is based on a negative result. The authors should consider using another approach to confirm this result. In vitro studies using immunoprecipitation for instance would be the best way to do so.
- The paper is entirely based on one approach (artificial tethering of overexpressed tagged histones) and one readout (microscopy). As mentioned above, the author should have tested another method to validate the key results.
- The artificial system of histone tethering is well described but extremely artificial. As a result, the conclusion for me, if it is true, is that it can happen, not that it happens under normal condition.
- If it happens, it is unknown whether it is a frequent process or if only a small proportion of the soluble histone pool is maintained as monomer. Therefore, it is impossible to estimate whether this finding is important for the chaperone-histone delivery pathway.

Additional points

- Over expression of histones do not affect cell morphology as pointed out by the authors but it affects DNA replication progression. The authors should monitor the impact of histone overexpression in their system as it will affect the pool of soluble histones, their import and the rate of incorporation of new histones.
- It seems that the choice of where to draw the "white line" is totally subjective. As a microscopy user myself, I am aware that from the same image you can observe very different patterns depending on where you draw the line. It should therefore not be used as a method to analyse co-localization (see additional comment below).
- The Pearson's correlation coefficient plotted are very low compare to the image showed (Pearson's coefficient correlation close to 0.8). The authors should present representative images, and not the best correlation image, it is misleading.
- In average, only few cells are analysed (n = 30 for most figures). I understand that using live microscopy, it is difficult to reach a high number of cells analysed. However, given the spread of the

Pearson's correlation coefficient, the authors would gain into robustness by increasing the number of cells analysed.

- In figures using PCNA-RFP, the authors always show cells in Mid-Late S phase. Why? Is the co-localization specific to this stage of S phase?

Referee #3:

This manuscript addresses the transport of newly synthesized histones from the cytosol to the nucleus for assembly into nucleosome. This is an important and in my opinion under-researched area, and thus the manuscript is a good contribution to the field. A whole network of histone-binding proteins (including importins) is thought to shuttle newly synthesized H3-H4 as a heterodimer. The authors show that fractionation and in vivo visualization of the various histone chaperones gives contradictory results. For example, they show that CAF1P160, which is nuclear, also appears interaction the cytoplasm after fractionation, possibly due to leakage of nuclear components. As such, tracking the rapid transport of proteins into the nucleus is challenging. The authors have developed a clever approach termed RAPID-release, where histones are first captured on the outer membrane, then released with a protease. This allows the authors to specifically monitor the fate of nascent cytosolic histones, once released from their tether. The caveat here is of course that the system is necessarily very artificial: the tethering of the histone, its release (how many amino acid does the protease leave behind?) and the attachment of the label. It is not obvious why the authors ignore the fact that C-terminal tethering, especially of H4, but also of H3, is potentially disruptive for nucleosome structure (even if the effects of a C-terminal tether on transport and assembly factor function per se might be more moderate than for N-terminal tethering. As such, it cannot be assumed that the tagged histone (especially H4), once it is in the nucleus, is stably incorporated into chromatin. Fig. 3E attempts to address this issue, but is not entirely convincing, as the tagged histone could aggregate on the mitotic chromosome. I would feel better if the authors could show MNase ladders with the fluorescent histone in it (or is the intensity not enough?). The results in Fig. 6 with the histone mutant somewhat alleviate that concern, as this describes an H3 mutant that clearly behaves differently.

The surprising finding here is that histones H3 and H4 exist as monomers in the cytosol and do not form the heterodimers that were thought to be obligatory. This is based largely on negative results (i.e. the absence of histone chaperones, or of the other histone partner), which could be due to the presence of the tags or because of tethering. Similarly, the forced histone chaperone interactions could be just that, 'forced' and thus unclear whether this is physiological.

The data showing colocalization with various importins is good. however, a control of a OMM-tethered protein that has no business interacting with any of the importins should be added.

The authors should at least address the potential pitfalls from the artificial system.

Fig. 2A: all of the abbreviations have to be explained in the figure legend. The figure, although well-intentioned is unintelligible to me.

Fig. 6A isnt very helpful in illustrating the role of the mutated residues.

Response to reviewers

Ref #1:

A method is employed to force nascent histones into the cytoplasm by tethering them to the outer membrane of mitochondria. Then, by an ingenious system that was previously developed, histones can be released from the outer mitochondrial membrane using a rapamycin inducible dimerization and protease activation system. This results in rapid nuclear import of the histones. The authors show that the system works and results in rapid translocation of histones into the nucleus upon Rapamycin induced cleavage. They go on to show that the degree of histone import correlates with the degree of Rapamycin dependent cleavage and that released histone are incorporated into chromatin in S phase.

The authors utilize this system to determine to what extent the known histone chaperones NASP, ASF1, HAT1 and RbAp46 associate with histone H3 and H4 while kept in the cytoplasm. They conclude that no such interaction takes place in the cytoplasm but only in the nucleus. They argue that prior data suggesting that such a prenucleosomal complex does exist in the cytoplasm is due to biochemical fractionation artifacts. When fractionating cells, NASP, RbAp46, HAT1, ASF1A and CAFp60 can all be found in the cytosolic pool but they authors argue that this is likely a soluble nuclear pool that is released into the cytoplasmic fraction. The experiments in figure 1 support this claim. Contrary to the histone chaperones, they find the nuclear receptors IPO4, IPO11 and KPNB1 to be associated with histones that are artificially tethered in the cytoplasm. Therefore, contrary to the currently literature, duly cited in this manuscript, the authors suggest that histone H3 and H4 do not form a heterodimer in the cytoplasm and do not interact with the canonical chaperones until they reach the nucleus.

The paper presents a series of well executed experiments and controls. The RAPID system, although artificial, allows the authors to arrest the very fast dynamics of nascent histone transfer to the nucleus, not possible with other techniques. The idea that histones complex with their cognate chaperones only in the nucleus is novel and relevant for the field. In my view this work should be published provided a few key issues are resolved, outlined below.

There is the potential caveat that there is a small fraction of histone chaperones present in the cytoplasm that goes beyond detection. The authors provided two sets of experiments indicating that this is unlikely:

1) To determine whether histone chaperones can, in principle, interact with cytoplasmic histones, mutant chaperones lacking an NLS or gaining an NES are expressed. These do interact with cytoplasmically tethered histones.

To test whether this interaction is relevant, the authors use FRAP experiments in Figure 5D-E as a measure of the strength of the interaction which they show is weak compared to e.g. important IPO4. In principle, this is a good approach to demonstrate that histone chaperones are unlikely to interact with histones in the cytoplasm. However, in these experiments RbAp46 with a NES or sNASP lacking an NLS is broadly localized to the cytoplasm and it is not clear whether the FRAPed pool represents the pool bound to H4 or H3.1, respectively. A control should be included to show that the localization of these mutant chaperones is histone dependent by expressing them in cells lacking the tethered histones.

We control for the specificity of forced cytosolic chaperones to mitochondria in Figure 4E through the tethering of eGFP alone. Both sNASP Δ NLS and the RbAp46-NES constructs specifically recruited to H3 and H4, respectively, but not to EGFP alone. As the same constructs were used in the FRAP experiment, this appears to represent the control the reviewer was asking for.

2) Another critical experiment that the authors provide as evidence that the prenucleosomal complexes in the cytoplasm are different from the ones in the nucleus is shown in figure 6A-E. The authors claim that H3 and H4 are monomeric in the cytoplasm by using a mutant form of H3 that is predicted to not interact with H4. It is clear that this mutant does not incorporate into chromatin yet is still translocated to the nucleus. However, whether it can still interact with H4 is not tested. This should be demonstrated.

The reviewer highlights an important control. The mutated H3 could fold with H4, but not be incorporated into chromatin. We have attempted to address this by immunoprecipitating the soluble pool of H3-EGFP soon after its release from the cytosol to check for folding with H4. However, due to the limited material available for pulldown and the rapid incorporation of non-mutated H3 into chromatin, we couldn't gain enough material to quantify the amount of interacting H4 in the wild-type. As this measurement is necessary for demonstrating the lack of H4 bound to mutant H3, which in turn

constitutes the control the reviewer is asking for, we decided to remove this experiment from the manuscript. We have replaced the figure with the new finding from quantitative IPs that sNASP interacts with monomeric H3 in the nucleus, which we believe provides stronger evidence using an alternative approach to lend more support for the conclusion that histones can be imported as monomers, and that a stable monomeric pool of H3 can exist in the nucleus. Thus, the major conclusion of the manuscript still remains.

Further, the inability of H3 to co-recruit H4 to the mitochondria (and visa versa) (figures 4C and EV3A) is taken as evidence that H3 and H4 are monomeric in the cytoplasm. However, it is also possible that the non-overexpressed partner is simply too low in abundance in the cytoplasm to detect. In the case of expressing mutant cytoplasmic histone chaperones, do the authors see co-recruitment of H3 to H4 or visa versa? This would control for the ability to detect these histones, if present.

The reviewer highlights that the lack of endogenous binding partner could be due to low abundance of the non-overexpressed histone. This certainly could be the case, however, as histones are one of the most abundant cellular proteins, and as the over-expressed histones are incorporated rapidly into chromatin upon release from the cytosolic tether (suggesting a significant pool of endogenous histone partner) (Figure 3), we would argue that this is not the case. The tethered histones themselves are detected well above background, offering some level of control.

There could be a small fraction of endogenous partner that is not detectable above background, as this and other reviewers point out. This then leads to the question of whether the finding using tethered histones is relevant to normal cellular circumstances. i.e. histones *can* be imported as monomers, but not necessarily *are* imported as monomers. We don't think that this will be easily answered by performing more cytosolic tethering experiments, so endeavoured to use a different approach looking at the stoichiometries of H3 and H4 associated with the core chaperoning machinery (Figure 6).

Manuscript structure:

The presentation of the discrepancy of nuclear fractionation versus imaging of histone chaperons is confusing. In the abstract this is presented as one of the final observations while in the text the authors start by outlining this discrepancy as a starting point of their work. It would help if this would be streamlined. The first paragraph is entitled: "Mislocalization of histone chaperones during fractionation necessitates an in vivo pulse labelling system" It is not clear from the paragraph that the mislocalization during fractionation necessitates this system. The authors simply explain why chaperones "appear" cytosolic while being in fact nuclear but they should spell out more clearly how their RAPID system can resolve this issue.

We had originally thought to present the fractionation experiment later in the manuscript, but came to the conclusion that it somewhat justifies the proceeding experiments, and so moved it to the front. The reviewer astutely picks up a conflict in the MS structure that shows this. The abstract has been rewritten to accommodate this and accommodate the new data displayed in Figure 6.

Ref #2:

This reviewer thought it was important to add more support for the conclusion that H3 and H4 are imported as monomers rather than heterodimers using additional approaches other than the trapping of histones in the cytosol through tethering mechanisms. This is very difficult to achieve as the cytosolic phase of histones is extremely short lived. Indeed, the tethering approach was developed due to these experimental difficulties. However, it is true that the major conclusion of our work relied in most part on such artificial tethering techniques and on primarily negative results. We have taken onboard the reviewers concerns, and in the revised manuscript provide new data independent of histone tagging or tethering that establishes a soluble pool of monomeric H3 bound to the chaperone sNASP, as detailed below. As we demonstrate that sNASP is nuclear in its sub-cellular location this offers strong support for a biological role of monomeric histones in the nucleus and lends further support to the major conclusion of the manuscript that histones H3 and H4 are imported as monomers rather than heterodimers.

In this manuscript, Apta-Smith and Bowman test the possibility that histone H3.1 and H4 can be transferred into the nucleus as monomers. To this end they have developed an artificial tethering system of overexpressed monomeric tagged histones to mitochondria. Upon rapamycin treatment, tagged histones are rereleased, transferred to the nucleus and incorporated behind replication forks. The key findings are: 1) Endogenous member of the H3.1-H4 dimer (Endogenous H4 for H3.1-tag, and

endogenous H3.1 for H4-tag) cannot be detected, suggesting that H3.1 and H4 can exist as monomers in the cytoplasm (Figure 4C). 2) Histones can be imported in the nucleus as monomer but not assembled onto newly replicated DNA (Figure 6C-E). I do not recommend this manuscript for revision for several reasons, listed below:

Main concerns

- The first finding is based on a negative result. The authors should consider using another approach to confirm this result. *In vitro* studies using immunoprecipitation for instance would be the best way to do so.
- The paper is entirely based on one approach (artificial tethering of overexpressed tagged histones) and one readout (microscopy). As mentioned above, the author should have tested another method to validate the key results.
- The artificial system of histone tethering is well described but extremely artificial. As a result, the conclusion for me, if it is true, is that it can happen, not that it happens under normal condition.
- If it happens, it is unknown whether it is a frequent process or if only a small proportion of the soluble histone pool is maintained as monomer. Therefore, it is impossible to estimate whether this finding is important for the chaperone-histone delivery pathway.

We thank the reviewer for highlighting these valid points, and the helpful suggestions they have made to address them. We agree that the manuscript would be much stronger if a second approach providing positive data could be employed without relying on tethering techniques to trap the transient cytosolic phase.

To this end we have pursued an *in vitro* pull-down analysis, as suggested by the reviewer, of the three core components of the H3-H4 chaperoning pathway – NASP, ASF1 and the HAT1 complex (HAT1-RbAp46). The rationale for this was that if H3 and H4 are imported as monomers, one could expect a stable monomeric pool of H3 and H4 in the nucleus, and this is most likely to be bound to one of the core chaperones that bind histones early on in the pathway. Whilst IPs of H3 and ASF1 are prevalent in the literature, little quantitative analysis of the ratios of H3 and H4 associated with different chaperones has been performed.

Using EGFP-fusions, we performed immunoprecipitation experiments and quantified the ratio of H3 to H4 associated with NASP, ASF1 and HAT1. HAT1 was used as a representative of the HAT1 complex (HAT1-RbAp46) rather than RbAp46 as RbAp46 is present in other chromatin complexes, such as HDACs, the NuRD complex and the PRC2 complex. Whilst ASF1 associated with a stoichiometric amount of H3 and H4, as is well documented, the chaperone NASP associated with a large excess of H3 over H4. As we demonstrate that NASP is nuclear, both in its endogenously expressed form (Figure 1) and as an EGFP fusion (Figure 6 & Figure EV1B), we take this finding as evidence for the existence of a stable monomeric H3 pool in the nucleus. The most obvious source for this monomeric pool would be newly synthesised histones, leading back to the major conclusion of the original manuscript, which is that H3 is rapidly imported as a monomer after synthesis. Furthermore, the presence of monomeric H3 bound to sNASP would fit well with a recent observation of NASP's H3 C-terminal binding site, and the observation that NASP can stably interact with monomeric H3 *in vitro* but not monomeric H4 (Bowman, Koide *et al.*, 2017, Bowman, Lercher *et al.*, 2016). Whilst we could detect both H3 and H4 associated with HAT1 by mass spectrometry, and whilst the H4 appeared to be in excess of H3 through Coomassie staining, the levels of histone were too low to make an accurate measurement. Thus, whether a soluble pool of monomeric H4 exists bound to a different chaperone is still an open question. Data from these immunoprecipitation experiments is presented in a new Figure 6 in the revised manuscript.

We hope that this additional quantitative IP analysis has gone some way in addressing the main concerns of the reviewer in (1) presenting a positive result, (2) providing a second approach to validate the key finding, (3) avoiding artificial cytosolic tethering and (4) giving an indication that a significant proportion of histone (H3) is in a stable monomeric state (bound to sNASP). We are interested in whether the reviewer finds these new data in support of our original conclusions.

Additional points

- Over expression of histones do not affect cell morphology as pointed out by the authors but it affects DNA replication progression. The authors should monitor the impact of histone overexpression in their system as it will affect the pool of soluble histones, their import and the rate of incorporation of new histones.

The reviewer is correct in pointing out that overexpression of *soluble* histones may affect the size of the endogenous pool and have effects on the rate at which histones are incorporated. As the measured

nuclear import rate in our system was shown to be limited by the kinetics of protease cleavage, we don't believe that the import of histones, the main focus of this current manuscript will be much affected by the pool of tethered substrate.

- It seems that the choice of where to draw the "white line" is totally subjective. As a microscopy user myself, I am aware that from the same image you can observe very different patterns depending on where you draw the line. It should therefore not be used as a method to analyse co-localization (see additional comment below).

We have carried out a new analysis of the colocalising data using PCC of background-corrected single Z-slices. This is detailed in the material and methods sections, with the major conclusion remaining the same: tethered H3 and H4 are monomeric and do not associate with the core chaperones NASP, ASF1, RbAp46 or HAT1, but do associate with IPO4 and, in the case of H4, KPNB1. In essence, using a single Z slice rather than a maximum intensity projection, and a 20 pixel rolling ball correction to smooth the gradient of fluorescence across the cell, whilst retaining features of the mitochondrial network, limited the background colocalization, as seen from the tethered EGFP alone. This is depicted more clearly in the main figures and also in the Expanded View Figure 3.

- The Pearson's correlation coefficient plotted are very low compared to the image showed (Pearson's coefficient correlation close to 0.8). The authors should present representative images, and not the best correlation image, it is misleading.

A full panel of sample images from one of the antibodies used (anti-H3) is now displayed along with the 2D histogram from each channel. The R-value of the images displayed is given in the inset so that the reader can judge the representation compared to the dataset.

- In average, only few cells are analysed (n = 30 for most figures). I understand that using live microscopy, it is difficult to reach a high number of cells analysed. However, given the spread of the Pearson's correlation coefficient, the authors would gain in robustness by increasing the number of cells analysed.

Analysing more cells would certainly reduce the P-values, although with the new analysis of the data the distribution is tighter around the mean, and thus we don't believe it would contribute meaningfully to the conclusions made from the data.

- In figures using PCNA-RFP, the authors always show cells in Mid-Late S phase. Why? Is the co-localization specific to this stage of S phase?

Using the RAPID-release system histones are incorporated into chromatin in all stages of S-phase. We found that histones released in G1 would sit in a soluble pool for a number of hours, and then could be incorporated into chromatin when S-phase started. As we were using asynchronous populations, to ensure that S-phase had commenced when the histones were released we only counted cells in which the archetypal peripheral pattern of mid-late replication foci was present. We could then be sure that the point at which histone were released (up to 5 hours previous) the cells had already entered S-phase.

Ref #3:

This manuscript addresses the transport of newly synthesized histones from the cytosol to the nucleus for assembly into nucleosome. This is an important and in my opinion under-researched area, and thus the manuscript is a good contribution to the field. A whole network of histone-binding proteins (including importins) is thought to shuttle newly synthesized H3-H4 as a heterodimer. The authors show that fractionation and in vivo visualization of the various histone chaperones gives contradictory results. For example, they show that CAF1P160, which is nuclear, also appears to interact in the cytoplasm after fractionation, possibly due to leakage of nuclear components.

As such, tracking the rapid transport of proteins into the nucleus is challenging. The authors have developed a clever approach termed RAPID-release, where histones are first captured on the outer membrane, then released with a protease. This allows the authors to specifically monitor the fate of nascent cytosolic histones, once released from their tether. The caveat here is of course that the system is necessarily very artificial: the tethering of the histone, its release (how many amino acid does the protease leave behind?) and the attachment of the label.

The amino acid sequences of the constructs are provided in the Expanded View, with domains from the fusion proteins marked. The cleavage site is C-terminal to the GFP, therefore the remaining amino

acids are not juxtaposed to the histone, which is N-terminal of the GFP, and therefore would like have little impact. The attachments of the labels can also be extracted from this data.

It is not obvious why the authors ignore the fact that C-terminal tethering, especially of H4, but also of H3, is potentially disruptive for nucleosome structure (even if the effects of a C-terminal tether on transport and assembly factor function per se might be more moderate than for N-terminal tethering. As such, it cannot be assumed that the tagged histone (especially H4), once it is in the nucleus, is stably incorporated into chromatin. Fig. 3E attempts to address this issue, but is not entirely convincing, as the tagged histone could aggregate on the mitotic chromosome. I would feel better if the authors could show MNase ladders with the fluorescent histone in it (or is the intensity not enough?). The results in Fig. 6 with the histone mutant somewhat alleviate that concern, as this describes an H3 mutant that clearly behaves differently.

Published evidence, in particular a seminal study by Kimura & Cook, 2001, (Kimura & Cook, 2001) would suggest that C-terminal tagging of histones H3 and H4 with GFP has minimal effect on their incorporation into chromatin and their turnover once deposited, whereas N-terminal tagging is highly disruptive. This is in compliance with our data where we show that C-terminally tagged H3 & H4 once released are incorporated into chromatin at sites of active replication. If the histones were aggregating on DNA one may expect to see a more homogenous pattern throughout the nucleus rather than enrichment at centres of active replication.

The surprising finding here is that histones H3 and H4 exist as monomers in the cytosol and do not form the heterodimers that were thought to be obligatory. This is based largely on negative results (i.e. the absence of histone chaperones, or of the other histone partner), which could be due to the presence of the tags or because of tethering. Similarly, the forced histone chaperone interactions could be just that, 'forced' and thus unclear whether this is physiological.

The reviewer is correct in highlighting that the major conclusion of this study is necessarily supported by negative results, i.e. the lack of detectable interaction, upon which it is always difficult to build definite statements. The reviewer also alludes to the presence of tags and the artificial nature of the tethering approach employed. To address these concerns, we have added new data to the manuscript in which we have pursued a quantitative immunoprecipitation approach to demonstrate the existence of monomeric H3 in the nucleus, associated with the chaperone sNASP, which has not relied on the tagging or tethering of histones (revised figure 6). The presence of monomeric H3 in the nucleus is supportive of a model in which histones H3 and H4 can be imported as monomeric, rather than dimeric, units. For more details please see the above response to reviewer 2 who highlighted the same concern.

The data showing colocalization with various importins is good. however, a control of a OMM-tethered protein that has no business interacting with any of the importins should be added.

The authors should at least address the potential pitfalls from the artificial system.

Fig. 2A: all of the abbreviations have to be explained in the figure legend. The figure, although well-intentioned is unintelligible to me.

The abbreviations have been included in the legend. The figure has been expanded in an attempt to make it more understandable.

Fig. 6A isnt very helpful in illustrating the role of the mutated residues.

For reasons detailed above (reviewer 1) data related to figure 6A has been removed and replaced with a revised figure.

References

Bowman A, Koide A, Goodman JS, Colling ME, Zinne D, Koide S, Ladurner AG (2017) sNASP and ASF1A function through both competitive and compatible modes of histone binding. *Nucleic Acids Res* 45: 643-656

Bowman A, Lercher L, Singh HR, Zinne D, Timinszky G, Carlomagno T, Ladurner AG (2016) The histone chaperone sNASP binds a conserved peptide motif within the globular core of histone H3 through its TPR repeats. *Nucleic Acids Res* 44: 3105-17

Kimura H, Cook PR (2001) Kinetics of core histones in living human cells: little exchange of H3 and H4 and some rapid exchange of H2B. *J Cell Biol* 153: 1341-53

Thank you for submitting a revised version of your manuscript. It has now been seen by all three original referees and their comments are shown below.

As you will see, refs #1 and #3 find that their main criticisms have been sufficiently addressed and recommend the manuscript for publication. Ref #2, on the other hand, remains unconvinced that the data supports nuclear import of histone monomers. Given the support from the other two referees, we are still interested in publishing your manuscript in The EMBO Journal but I would ask you to submit a final revised version where you comment on the concerns from ref #2. In addition, please address the following editorial issues concerning text and figures:

REFeree REPORTS

Referee #1:

The authors have greatly improved their manuscript and have addressed my principal concerns. Improvements are the manner in which co-localization of mitochondrial-tethered histones and chaperones have been quantified in figures 4B and E and 5A.

A key concern of mine as well as reviewers 2 and 3 was the artificial nature of their RAPID-release system. While necessary and powerful, it raised some doubts as to whether the discovery of monomeric histone chaperone binding is a consequence of the engineering of the system. To address these concerns, the authors have now added a pulldown experiment of histone chaperones in order to quantify the relative ratio of H3 vs H4 that comes down with specific chaperones. They show adequate control for their ability to measure these ratios in EV5. These experiments reveal evidence that at least in the case of NASP, predominantly H3 is co-precipitated, suggesting H3 is bound not as a stoichiometric heterodimer with H4 but rather as a monomer, at least for some NASP complexes. This combined with their cytosolic tethering system provides good evidence that H3 and H4 can be imported as monomers and handed off to nuclear chaperones as such.

At this point, I believe all reasonable concerns have been addressed and I support publication of the paper.

Referee #2:

The existence of H3 monomer has been reported previously by gel filtration (Campos EI et al 2010 NSMB). The novelty of this paper was that H3 could be imported in the nucleus as a monomer. In my opinion the addition of Figure 6 strengthens the paper. However, the authors do their pulldowns in whole cell soluble extract, not in nuclear fraction (Figure 6B). Therefore, these results do not support the conclusion that H3 exist as a monomer in nucleus. The control the authors provide to justify using whole cell soluble extract instead of proper nuclear extract is done by immunofluorescence images (Figure 6A). This is 1) not quantitative, and 2) not a comparison of soluble cytoplasmic versus soluble nuclear, but cytoplasmic versus nuclear + chromatin! Therefore, I am not convinced that H3 monomer are imported in the nucleus and do not recommend this paper for publication.

Referee #3:

I am satisfied with the added experiment. While i would have wished for additional proof for this intriguing model, I think that the concept is of sufficiently high novelty and interest that it should be published in its present form.

Response to referees:

Referee #1:

The authors have greatly improved their manuscript and have addressed my principal concerns. Improvements are the manner in which co-localization of mitochondrial-tethered histones and chaperones have been quantified in figures 4B and E and 5A. A key concern of mine as well as reviewers 2 and 3 was the artificial nature of their RAPID-release system. While necessary and powerful, it raised some doubts as to whether the discovery of monomeric histone chaperone binding is a consequence of the engineering of the system. To address these concerns, the authors have now added a pulldown experiment of histone chaperones in order to quantify the relative ratio of H3 vs H4 that comes down with specific chaperones. They show adequate control for their ability to measure these ratios in EV5. These experiments reveal evidence that at least in the case of NASP, predominantly H3 is co-precipitated, suggesting H3 is bound not as a stoichiometric heterodimer with H4 but rather as a monomer, at least for some NASP complexes. This combined with their cytosolic tethering system provides good evidence that H3 and H4 can be imported as monomers and handed off to nuclear chaperones as such. At this point, I believe all reasonable concerns have been addressed and I support publication of the paper.

Referee #2:

The existence of H3 monomer has been reported previously by gel filtration (Campos EI et al 2010 NSMB). The novelty of this paper was that H3 could be imported in the nucleus as a monomer. In my opinion the addition of Figure 6 strengthens the paper. However, the authors do their pulldowns in whole cell soluble extract, not in nuclear fraction (Figure 6B). Therefore, these results do not support the conclusion that H3 exist as a monomer in nucleus. The control the authors provide to justify using whole cell soluble extract instead of proper nuclear extract is done by immunofluorescence images (Figure 6A). This is 1) not quantitative, and 2) not a comparison of soluble cytoplasmic versus soluble nuclear, but cytoplasmic versus nuclear + chromatin! Therefore, I am not convinced that H3 monomer are imported in the nucleus and do not recommend this paper for publication.

Whilst Campos et al 2010, NSMB, do isolate a monomeric H3, they propose a model in which NASP, and Hat1-Rbap46, associate with a folded H3-H4 dimer in the cytosol, and dissociate before its import into the nucleus. This is understandable as long as one assumes no leakage from soluble nuclear components occurs during the sub-cellular fractionation process. While this could be the case for manual enucleation, from which the xenopus homolog of NASP (N1/N2) was originally discovered (Bonner, 1975), and later characterised to interact specifically with H3 & H4 (Kleinschmidt & Franke, 1982), the use of detergents or osmosis for lysis clearly results in the rapid leakage of soluble nuclear proteins. We demonstrate this in Figure 1, additionally showing that Lamin A/C, the archetypal nuclear control protein, is a poor indicator for assessing fractionation of soluble proteins.

Manual enucleation is not possible when working with human cell lines and carrying out biochemical analysis, thus we call the extract obtained using standard NP40 lysis protocols 'whole cell soluble extract', indicating it contains both soluble cytosolic and soluble nuclear components. We would venture that a true soluble cytosolic, or a true soluble nuclear extract is not achievable using detergent based or osmolytic cell lysis. To validate the true location of the EGFP-tagged proteins we use in the pulldowns we assessed their location by fluorescence microscopy (Figure 6A). This was in agreement with our quantitative analysis of endogenous chaperone localisation (Figure 1B & C) demonstrating that NASP, ASF1 and HAT1 are nuclear. Therefore, to us, the simplest conclusion is that the excess of monomeric H3 bound to sNASP is nuclear, which would support the idea of histones H3 & H4 being imported as monomers and forming a heterodimer in the nucleus.

Referee #3:

I am satisfied with the added experiment. While i would have wished for additional proof for this intriguing model, I think that the concept is of sufficiently high novelty and interest that it should be published in its present form.

References

- Bonner WM (1975) Protein migration into nuclei. II. Frog oocyte nuclei accumulate a class of microinjected oocyte nuclear proteins and exclude a class of microinjected oocyte cytoplasmic proteins. *J Cell Biol* 64: 431-7
- Kleinschmidt JA, Franke WW (1982) Soluble acidic complexes containing histones H3 and H4 in nuclei of *Xenopus laevis* oocytes. *Cell* 29: 799-809

Andrew J. Bowman
EMBO Journal
EMBOJ-2017-98714R